# Nse5/6 inhibits the Smc5/6 ATPase and modulates DNA substrate binding

Michael Taschner[1] (ID), Jérôme Basquin[2,†], Barbara Steigenberger[2,3,4,†], Ingmar B Schäfer[2], Young-Min Soh[1], Claire Basquin[2], Esben Lorentzen[5] (ID), Markus Räschle[6], Richard A Scheltema[3,4] & Stephan Gruber[1,*] (ID)

## Abstract

Eukaryotic cells employ three SMC (structural maintenance of chromosomes) complexes to control DNA folding and topology. The Smc5/6 complex plays roles in DNA repair and in preventing the accumulation of deleterious DNA junctions. To elucidate how specific features of Smc5/6 govern these functions, we reconstituted the yeast holocomplex. We found that the Nse5/6 sub-complex strongly inhibited the Smc5/6 ATPase by preventing productive ATP binding. This inhibition was relieved by plasmid DNA binding but not by short linear DNA, while opposing effects were observed without Nse5/6. We uncovered two binding sites for Nse5/6 on Smc5/6, based on an Nse5/6 crystal structure and cross-linking mass spectrometry data. One binding site is located at the Smc5/6 arms and one at the heads, the latter likely exerting inhibitory effects on ATP hydrolysis. Cysteine cross-linking demonstrated that the interaction with Nse5/6 anchored the ATPase domains in a non-productive state, which was destabilized by ATP and DNA. Under similar conditions, the Nse4/3/1 module detached from the ATPase. Altogether, we show how DNA substrate selection is modulated by direct inhibition of the Smc5/6 ATPase by Nse5/6.

**Keywords** chromosome segregation; cohesion; condensin; loop extrusion; Smc5/6
**Subject Categories** DNA Replication, Recombination & Repair; Structural Biology
**The EMBO Journal (2021) 40: e107807**

## Introduction

Maintenance of chromosome structure and the faithful transmission of genetic information are essential processes in all domains of life orchestrated by the widely conserved structural maintenance of chromosomes (SMC) complexes [reviewed in (Yatskevich *et al*, 2019)]. The main function of these ATP-powered DNA-folding machines in bacteria is to prevent entanglement of newly replicated chromosomes to ensure their unperturbed segregation to daughter cells. Eukaryotic cells also need to prevent such entanglements during cell division, but additionally efficient sister chromatid cohesion, chromosome condensation and chromosome individualization need to be ensured. Three distinct eukaryotic SMC complexes (cohesin, condensin and Smc5/6) divide these tasks between them. Cohesin folds interphase chromosomes into defined domains to regulate gene expression (Szabo *et al*, 2019) and participates in DNA repair by homologous recombination (Litwin *et al*, 2018). It also holds sister chromatids together between S-phase and the onset of anaphase (Yatskevich *et al*, 2019). Condensin compacts and structures chromosomes in mitosis to promote sister chromatid resolution and disjunction in prometaphase and anaphase, respectively (Hirano, 2016). The molecular functions of the Smc5/6 complex are understood in less detail (Aragon, 2018). Several Smc5/6 genes were first identified in screens for DNA damage-sensitive mutants (Prakash & Prakash, 1977; Lehmann *et al*, 1995; McDonald *et al*, 2003; Onoda *et al*, 2004; Torres-Rosell *et al*, 2005a; Torres-Rosell *et al*, 2005b). Complete Smc5/6 loss of function leads to cell death associated with severe chromosome segregation defects during both mitotic and meiotic cell divisions (McDonald *et al*, 2003; Pebernard *et al*, 2004; Copsey *et al*, 2013; Xaver *et al*, 2013). Without Smc5/6, certain toxic DNA structures such as unresolved recombination intermediates, DNA intertwinings, or incompletely replicated chromosomal regions prevent proper chromosome segregation (Torres-Rosell *et al*, 2005b; Torres-Rosell *et al*, 2007; Kegel *et al*, 2011), especially at repeated DNA sequences such as the ribosomal DNA arrays (Peng *et al*, 2018). It is however unclear whether Smc5/6 prevents their formation or promotes their removal.

1 Department of Fundamental Microbiology (DMF), Faculty of Biology and Medicine (FBM), University of Lausanne (UNIL), Lausanne, Switzerland
2 Max Planck Institute of Biochemistry, Martinsried, Germany
3 Biomolecular Mass Spectrometry and Proteomics, Bijvoet Center for Biomolecular Research and Utrecht Institute for Pharmaceutical Sciences, Utrecht University, Utrecht, The Netherlands
4 Netherlands Proteomics Centre, Utrecht, The Netherlands
5 Department of Molecular Biology and Genetics, Aarhus University, Aarhus, Denmark
6 Molecular Genetics, University of Kaiserslautern, Kaiserslautern, Germany
*Corresponding author. Tel: +41 21 692 5601; E-mail: stephan.gruber@unil.ch
†These authors contributed equally to the work

At their core all SMC complexes have a dimer of SMC proteins, each of which contains a "hinge" domain that mediates SMC dimerization and connects via a long (35–50 nm) antiparallel coiled-coil "arm" to a globular ABC-type "head" domain with highly conserved motifs for ATP binding and hydrolysis (Hirano *et al*, 2001; Lammens *et al*, 2004; Hopfner, 2016). Two ATP molecules are sandwiched by residues of the Walker A and B motifs of one SMC subunit and the signature motif of the other SMC subunit. ATP hydrolysis by the SMC heads is essential for the function of all SMC complexes, but is rather slow (mostly < 1 ATP/s) compared to other ATPases. It is however stimulated in the presence of DNA substrates [reviewed in (Hassler *et al*, 2018)] (Fousteri & Lehmann, 2000). The ATP hydrolysis cycle involves major structural rearrangements within the SMC dimers, which have been delineated only in some detail, for example, for the prokaryotic Smc-ScpAB complex (Soh *et al*, 2015; Diebold-Durand *et al*, 2017; Burmann *et al*, 2019; Chapard *et al*, 2019; Vazquez Nunez *et al*, 2019; Lee *et al*, 2020). Briefly, in the absence of ATP the two SMC proteins are in a "juxtaposed" J-state with closely aligned arms and a clear rod-shaped appearance in electron micrographs (Diebold-Durand *et al*, 2017; Lee *et al*, 2020). The heads contact each other close to their signature motifs to form a structure that is incompatible with ATP hydrolysis. In the presence of ATP, the head domains rearrange to adopt the "ATP-engaged" E-state with sandwiched ATP molecules (Lammens *et al*, 2004; Diebold-Durand *et al*, 2017). This conformation is incompatible with the rod conformation. At the least, it opens the arms in the head-proximal area, yielding a more open ring-like complex that has been observed for cohesin by cryo-EM (Higashi *et al*, 2020; Shi *et al*, 2020) and characterized in Smc-ScpAB by electron paramagnetic resonance and cross-linking (Vazquez Nunez *et al*, 2021). Upon ATP hydrolysis, the heads disengage, the coiled coils zip back up, and the complex reverts back into the J-state. The presence of DNA presumably assists the complex with these structural transitions, leading to a positive effect on the ATP hydrolysis rate. Indeed, DNA binding sites on top of the engaged heads were described for several related complexes (Liu *et al*, 2016; Seifert *et al*, 2016; Vazquez Nunez *et al*, 2019; Higashi *et al*, 2020; Shi *et al*, 2020).

Another invariably conserved subunit, called "kleisin", asymmetrically bridges the two SMC proteins at their head domains to create a tripartite ring structure capable of entrapping DNA in its lumen (Haering *et al*, 2004; Palecek *et al*, 2006; Burmann *et al*, 2013; Gligoris *et al*, 2014; Wilhelm *et al*, 2015). Kleisin also serves as an attachment point for additional proteins from the KITE (Kleisin-Interacting Tandem winged-helix Element) or HAWK (HEAT-protein Associated With Kleisin) families (Palecek & Gruber, 2015; Wells *et al*, 2017) for which functions related to DNA substrate interactions and ATPase regulation are emerging (Zabrady *et al*, 2016; Kschonsak *et al*, 2017; Li *et al*, 2018; Vondrova *et al*, 2020).

A fully assembled SMC complex utilizes the energy released by ATP hydrolysis in two manners: (i) "Topological entrapment" of DNA molecules inside the tripartite SMC/kleisin ring after regulated opening of (an) entry/exit gate(s) is the main mechanism by which cohesin holds sister chromatids together (Gligoris *et al*, 2014). Similar activities have also been described for condensin (Cuylen *et al*, 2011; Cuylen *et al*, 2013), Smc5/6 (Kanno *et al*, 2015; Gutierrez-Escribano *et al*, 2020) and prokaryotic Smc-ScpAB (Wilhelm *et al*, 2015), albeit in less mechanistic detail. (ii) "Loop extrusion" refers to an active ATP-dependent DNA-folding process and is used by condensin to compact mitotic chromatids and by cohesin to shape interphase chromosomes (Yatskevich *et al*, 2019). Such a biochemical activity has so far been reconstituted *in vitro* for cohesin and condensin (Ganji *et al*, 2018; Davidson *et al*, 2019; Kim *et al*, 2019) but it may well be a conserved feature of all pro- and eukaryotic relatives.

Several features make the Smc5/6 complex with its two SMC proteins (Smc5 and Smc6) and six "Non-SMC Elements" (Nse1-6) profoundly different from cohesin and condensin (Haering & Gruber, 2016). The kleisin Nse4 is a comparatively small protein. It binds two interactors (Nse1 and Nse3) belonging to the KITE family rather than the HAWK family as in cohesin and condensin (Palecek *et al*, 2006; Doyle *et al*, 2010; Palecek & Gruber, 2015; Wells *et al*, 2017). The Nse2 subunit attaches to the coiled-coil arm of Smc5. It lacks known relatives in other SMC complexes (Duan *et al*, 2009a). Apart from the ATPase activity located in the Smc5 and Smc6 heads, the complex harbours a small ubiquitin-related modifier (SUMO) ligase activity in Nse2 (Andrews *et al*, 2005; Zhao & Blobel, 2005) as well as a ubiquitin ligase activity in Nse1 (Doyle *et al*, 2010). Mutants with disruptions in these enzymatic domains are viable but sensitive to DNA damage, and these activities are thus only required for non-essential functions of Smc5/6 (Andrews *et al*, 2005; Potts & Yu, 2005; Pebernard *et al*, 2008). Complete removal of any of these subunits is however lethal in yeast (McDonald *et al*, 2003).

The last two subunits, Nse5 and Nse6, form a stable heterodimer and have only very weak sequence similarity to their presumed vertebrate counterparts Slf1 and Slf2 (Raschle *et al*, 2015). Experiments on this sub-complex performed in budding and fission yeast have yielded several disparate results. While Nse5/6 has been reported to bind to the head-proximal region of the Smc5/6 hexamer in *S. pombe* (Palecek *et al*, 2006), it was shown to bind to the hinge domain in *S. cerevisiae* (Duan *et al*, 2009b). Recent mapping experiments with human Slf1/2 showed a binding mode similar to that observed in fission yeast (Adamus *et al*, 2020). While neither Nse5 nor Nse6 is essential in *S. pombe* (Pebernard *et al*, 2006), they are required for viability in *S. cerevisiae* even under unperturbed conditions (Zhao & Blobel, 2005; Aragon, 2018) (Fig 3D). Nse5/6 is involved in the DNA repair function of Smc5/6 (Pebernard *et al*, 2006; Bustard *et al*, 2012), potentially by working together with Nse2 in substrate SUMOylation (Bustard *et al*, 2016)). It also has a role in recruiting the Smc5/6 complex to DNA damage sites through an interaction between an N-terminal unstructured peptide in Nse6 and a multi-BRCT domain of Rtt107 (Leung *et al*, 2011; Wan *et al*, 2019). Single-molecule tracking recently suggested a function for Nse5/6 in chromosomal loading of Smc5/6 (Etheridge *et al*, 2021).

To elucidate the function of budding yeast Nse5/6, we determined the Nse5/6 crystal structure and investigated its interaction with the Smc5/6 core hexamer as well as its influence on the Smc5/6 ATPase. We found that Nse5/6 strongly inhibited Smc5/6 ATPase function by preventing productive ATP binding, likely by inducing a major rearrangement of the Smc5 and Smc6 head domains. Addition of plasmid DNA, but not of short linear molecules, robustly stimulated ATP hydrolysis by the holo-complex but not the core hexamer, thus suggesting that Nse5/6 modulates DNA substrate binding by the Smc5/6 ATPase. To elucidate the organization of the holo-complex, we performed cross-linking mass spectrometry experiments (XL-MS). The data revealed major conformational changes involving the Nse4/3/1 kleisin/KITE module upon ATP and DNA

binding. In summary, our experiments demonstrate that Nse5/6 is a key partner that associates with the Smc5/6 heads to allow salt-stable DNA binding and to modulate the ATP hydrolysis rate of Smc5/6 in response to DNA substrate binding.

# Results

## Reconstitution of the Smc5/6 octamer

Here, we focused on the biochemical and structural analysis of the "loader" Nse5/6 in the context of the Smc5/6 holo-complex. To do so, we separately reconstituted the Smc5/6 core hexamer of yeast origin and the corresponding Nse5/6 dimer by co-expression of subunits in *E. coli*. The hexamer was enriched by affinity purification using a Twin-Strep-tag on Smc6, while the dimer was purified using a His-tag on Nse5. The protein preparations were further polished by ion exchange chromatography and gel filtration. Both complexes eluted from an analytical gel filtration column as a single species with apparently stoichiometric subunit composition (Fig 1A). Mixing of the hexamer and dimer prior to gel filtration resulted in a shift to a smaller elution volume, indicating the formation of a stable Smc5/6 octamer.

## The architecture of Smc5/6

To determine the overall organization of the Smc5/6 octamer, we first performed lysine-specific XL-MS to identify residues in close proximity. Using the short-spacer, enrichable, lysine-reactive cross-linking reagent PhoX (Steigenberger *et al*, 2019), we detected 98 intra-subunit cross-links (intra-links) as well as 64 inter-subunit cross-links (inter-links) on the reconstituted Smc5/6 octamer in a buffer containing 250 mM NaCl (Fig 1B). Additional cross-links were detected with another buffer (see below). The pattern of intra-links in Smc5 and Smc6 was generally in good agreement with what is expected for SMC proteins folded at the central hinge domain (Fig 1C). A sizeable fraction of the inter-links were located between the coiled coils of Smc5 and Smc6, strongly supporting the notion of the co-alignment of the Smc5 and Smc6 coiled coils from the hinge to the head domains in a juxtaposed conformation (J-state) (Fig 1C). Under these conditions, only one intra-subunit cross-link (in red colour) deviated from this Smc5/6 rod pattern, which could represent a false positive hit as the data analysis was cut off at 1% false positive rate. The cross-link may also have formed between two Smc5 proteins. Similar cross-linking was performed in a different buffer yielding largely comparable results that are discussed below. The cross-links of Smc5 and Smc6 residues to the other subunits, including Nse5 and Nse6, are also discussed further below, in the context of the Nse5/6 crystal structure.

To investigate the putative folding of Smc5/6 arms by a complementary method, we analysed yeast Smc5/6 hexamers by cryogenic electron microscopy. We obtained 2D class averages that showed highly elongated particles with a length of about 45 nm, consistent with fully extended and co-aligned Smc5/6 arms (Fig 1D, images 1 and 2). An additional density along the arms was noticeable at the anticipated position for the Nse2 subunit. Near the head domains, individual arms were recognizable, often in a slightly bent form as expected from a disruption in the coiled-coil structure near the

heads, called the SMC joint (Diebold-Durand *et al*, 2017). Of note, some class averages included dimeric forms of the hexamer in a head-to-head configuration (Fig 1D, images 3 and 4). The relevance of these dimers remains to be determined.

## Crystal structure of Nse5/6

Structural information on the Nse5/6 complex is currently limited to the disordered N-terminus of Nse6 that interacts with Rtt107 (Wan *et al*, 2019) (Fig 2A). Compounding this, there are conflicting reports in the literature on the presence of α-helical HEAT repeats in Nse6 and its putative relatedness to human Slf2, a subunit of the Smc5/6 associated Slf1/Slf2 complex (Pebernard *et al*, 2006; Raschle *et al*, 2015; Adamus *et al*, 2020; Yu *et al*, 2021). Limited proteolysis of purified Nse5/6 (Fig 2B) led to the design of two N-terminally truncated Nse6 fragments (154-C and 177-C) which were easily co-purified with Nse5 indicating a stable interaction with Nse5. The Nse5/Nse6(177-C) complex yielded selenomethionine substituted protein crystals that diffracted to 3.3 Å resolution and provided experimental phase information to compute an electron density of high enough quality to build a structural model (Figs 2C and EV1) (Table 1). In the electron density map, residues 284–464 of Nse6 were visible, corresponding to the domain with predicted α-helical secondary structure (Fig 2A). It indeed folds into 11 α-helices, arranged in an overall crescent-shaped structure (Figs 2C and EV1A). We did not observe reliable density for Nse6 residues 177–283, indicating flexibility in this region in the crystal. For Nse5, we observed clear density for 16 α-helices formed by residues 2–518 (Figs 2C and EV1B), with some of the helices being connected by apparently extended loops (147–197, 290–340 and 430–491), for which we only detected poor density or none at all.

The Nse5/6 interaction interface covers around 1,300 Å² in the crystal (Fig 2D), as calculated by the PISA server (Krissinel & Henrick, 2007). On Nse6, it mainly involves residues located in helices α3, α6, α9 and α11 together forming a concave interaction surface with a clear pattern of sequence conservation (Fig EV1A). Its pronounced hydrophobic area packs tightly against Nse5 residues forming helices α2 and α5, as well as a loop connecting helices α4 and α5. Sidechains around this hydrophobic pocket engage in further polar contacts. Additional, albeit less extended and conserved, contacts exist between residues in helix α12 as well as the preceding loop in Nse5 with helices α1 and α3 of Nse6 (Fig EV1C). The unmodelled Nse6 sequences further towards the N-terminus might strengthen this part of the interface. Due to the large interaction surface and the fact that neither Nse6 nor Nse5 could be produced in isolation, we did not attempt to disrupt this interface by mutagenesis. Such mutations, however, have been reported based on a related structure obtained by cryo-electron microscopy recently (Yu *et al*, 2021). To evaluate our crystal structure using a different approach, we mutated H368 in Nse6 as well as G56 in Nse5 to cysteines. The resulting cysteine pair should be ideally positioned for cross-linking using the thiol-specific cross-linker BMOE. We indeed observed robust cross-linking for the cysteine mutant but not the wild-type complex (Fig EV1D), showing that our model allowed us to make accurate predictions.

An extended surface area jointly formed by Nse5 and Nse6 residues displayed clear evolutionary conservation (Fig 2E). This surface is unlikely to bind to unmodelled N-terminal Nse6 residues

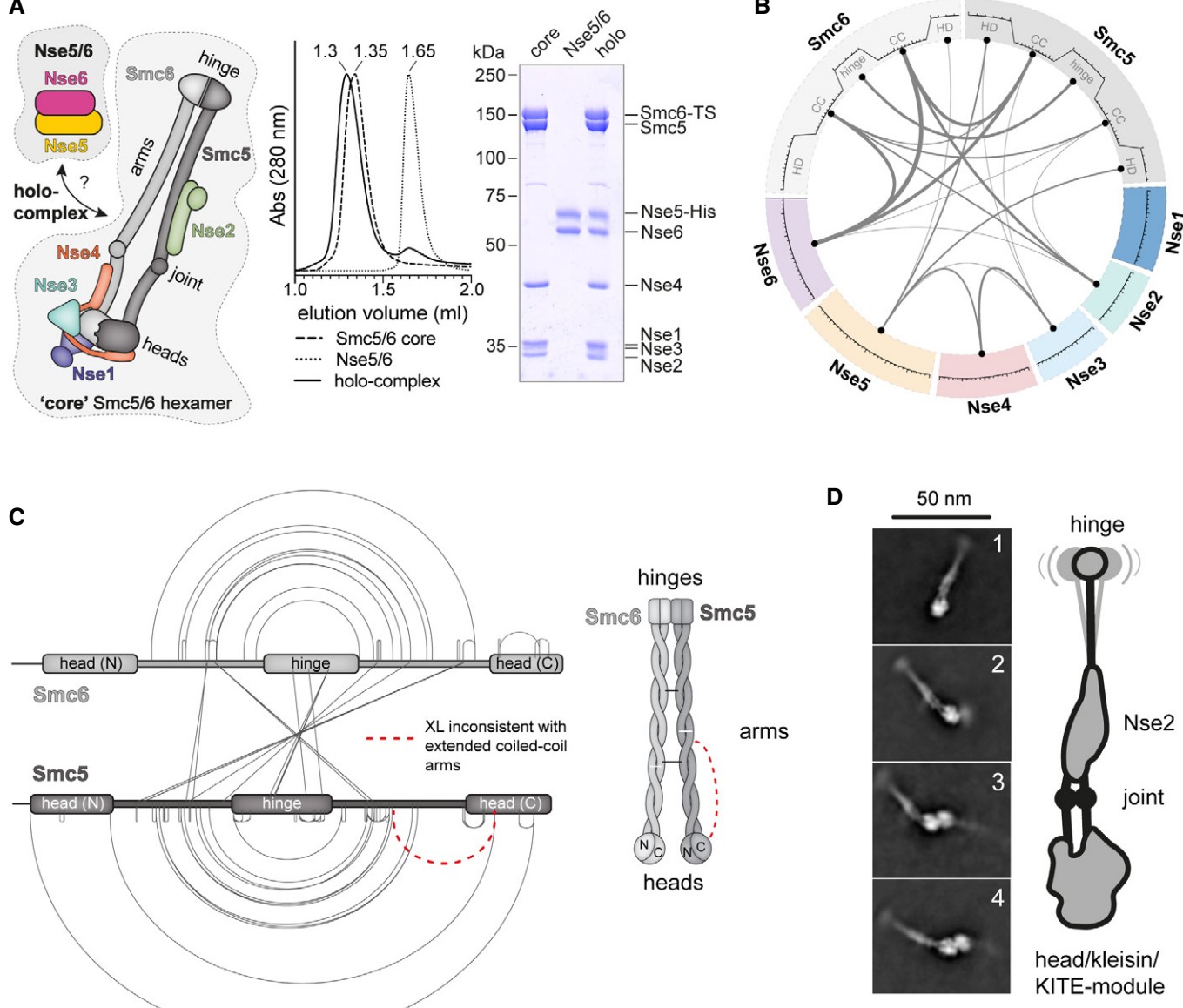

**Figure 1. Molecular architecture of yeast Smc5/6 complexes determined by cross-linking mass spectrometry (XL-MS) and cryo-electron microscopy (cryo-EM).**

A   Reconstitution of Smc5/6 and Nse5/6 complexes. *Left panel*, schematic depiction of the composition and organization of the yeast Smc5/6 "core" hexamer and the Nse5/6 dimer. *Middle panel*, elution profiles for analytical gel filtration (Superose 6 Increase 3.2/300) of the Smc5/6 core hexamer, the Nse5/6 dimer and a holo-complex obtained by mixing of dimer and hexamer. Measured by absorption at 280 nm. *Right panel*, peak fractions were analysed by SDS–PAGE and Coomassie Brilliant Blue staining.

B   Circular representation of lysine–lysine inter-subunit cross-links identified by mass spectrometry (XL-MS) in buffer containing 250 mM NaCl. For simplicity, cross-links between proteins (or domains) are grouped, and the thickness of the lines indicates the total number of cross-links of this particular type. Smc5 and Smc6 proteins are divided into N- and C-terminal head domains (HD) and coiled-coil arms (CC) as well as the central hinge domain. For a full representation of individual inter-subunit as well as intra-subunit cross-links, see Dataset EV1.

C   Cross-links obtained within and between Smc5 and Smc6 subunits of the Smc5/6 octamer. A cross-link that did not match to the elongated rod-shaped particle is displayed by a dashed line in red colours. The cartoon on the right shows the dimer of folded Smc5 and Smc6 proteins, with examples of intra- and inter-links within the coiled-coil arms indicated with white and black lines, respectively.

D   Selected 2D class averages obtained by cryo-electron microscopy of the yeast Smc5/6 hexamer (left images). Representative classes with dimers of Smc5/6 hexamers are displayed (images 3 and 4). Emerging details are indicated schematically (*right panel*).

Source data are available online for this figure.

**Table 1. Data collection and refinement statistics.**

| | Nse5/Nse6(177-C)* |
|---|---|
| Wavelength | 0.9792 |
| Resolution range | 19.85–3.293 (3.41–3.293) |
| Space group | P 21 21 2 |
| Unit cell | 99.116 147.368 74.444 90 90 90 |
| Total reflections | 91905 (7109) |
| Unique reflections | 16988 (1639) |
| Multiplicity | 5.4 (4.3) |
| Completeness (%) | 99.09 (98.38) |
| Mean I/sigma(I) | 16.77 (1.98) |
| Wilson B-factor | 110.80 |
| R-merge | 0.07468 (0.6013) |
| R-meas | 0.08246 (0.6831) |
| R-pim | 0.03437 (0.318) |
| CC1/2 | 0.999 (0.858) |
| CC* | 1 (0.961) |
| Reflections used in refinement | 16972 (1636) |
| Reflections used for R-free | 1700 (163) |
| R-work | 0.2971 (0.3900) |
| R-free | 0.3075 (0.4063) |
| CC(work) | 0.885 (0.764) |
| CC(free) | 0.871 (0.728) |
| Macromolecules | 4934 |
| Protein residues | 658 |
| RMS(bonds) | 0.005 |
| RMS(angles) | 1.09 |
| Ramachandran favoured (%) | 98.59 |
| Ramachandran allowed (%) | 1.41 |
| Ramachandran outliers (%) | 0.00 |

Statistics for the highest-resolution shell are shown in parentheses.

as these regions would be located on the opposite side of the structure. The surface area might be responsible for a putative interaction between Nse5/6 and the Smc5/6 core hexamer (see below).

We next searched for structural homologs using the DALI server (Holm, 2020). Searches with the complete Nse6(284–454) model in the Protein Data Bank revealed weak overall similarity (root mean square displacement (rmsd) > 4 Å) to a number of proteins, some of which contained HEAT repeats, with the best structural overlap found for the 4 C-terminal α-helices. Searches for matches for only these 4 C-terminal helices improved the rmsd values to a range of 2.0–2.6 Å for the best hits including eIF4G, exportin-1 and PTAR1, all containing HEAT repeats (Marcotrigiano *et al*, 2001; Dong *et al*, 2009; Kuchay *et al*, 2019). Structural similarity was clearly visible upon manual superpositioning of the corresponding regions (Fig EV1E, left). Similar DALI searches for the Nse5 model did not reveal significant hits, suggesting that the overall fold was not observed in other proteins so far. Restricting the search for matches to only the isolated N-terminal region involved in the main interaction with Nse6 (Fig 2D), uncovered several hits with rather low rmsd values in the range of 2.9–4 Å. Several of these again contained HEAT repeats, and a conserved overall positioning of individual helices was observed (Fig EV1E, right). The HEAT repeat-type organization in Nse5 and Nse6 however is limited to the immediate interface.

It remains unclear whether Nse5 and Nse6 have a common origin with the HEAT repeat subunits of cohesin and condensin. The Nse5/6 structure does not display the characteristic hook-shaped architecture of many HEAT repeat proteins, including the HAWK subunits of cohesin and condensin. We speculate, however, that Nse5 and Nse6 have evolved from larger ancestral proteins—being reminiscent of Slf1 and Slf2—by the loss of HEAT repeats. While this work was under review an independent study reported the cryo-EM structure of the budding yeast Nse5/6 complex (Yu *et al*, 2021), which is in excellent agreement with the crystal structure presented here (rmsd 0.829 Å; see Fig EV1F).

**Mapping of Nse5/6 contact points on the Smc5/6 hexamer**

Next, we mapped the binding sites for the Smc5/6 hexamer on the Nse5/6 dimer by performing pulldown assays using truncated Nse5/6 constructs. The crystallized complex containing the truncated Nse6 fragment (177-C) bound to Nse5 failed to stably associate with the Smc5/6 hexamer in these assays (Fig 3A). An N-terminal Nse6 fragment (1–179) fused to a CPD-His tag was also readily purified. It bound well to immobilized Smc5/6 hexamer (Fig 3A) but failed to co-purify with Nse5. Similarly, a Nse6 (86–179)-CPD-His fragment interacted with the Smc5/6 hexamer in pulldown assays (Fig EV2C). The N- and C-terminal sequences of Nse6 are therefore sufficient for binding to the Smc5/6 hexamer and to

**Figure 2. A Nse5/6 co-crystal structure.**

A  Domain organization of Nse5 and Nse6 proteins. Putative domain boundaries identified by secondary structure prediction and limited proteolysis are denoted by arrowheads, in black colours when producing stable fragments, otherwise in grey colours.

B  Limited proteolysis of a purified preparation of the Nse5/Nse6 complex by trypsin. Samples taken at different time points were analysed by SDS–PAGE and Coomassie staining. Selected stable fragments identified by mass spectrometry are indicated.

C  Front and back view of the Nse5/Nse6 co-crystal structure in cartoon representation. Structural elements of Nse5 and Nse6 are displayed in orange and purple colours, respectively. α-helices are labelled.

D  Conservation of residues at the Nse5/6 binding interface. Interaction surface of Nse6 (*left panel*) and Nse5 (*right panel*). For orientation, the secondary structure at the interface is displayed (*middle panel*). Colour code for residue conservation is given at the bottom of the panel.

E  A conserved surface area (*bottom panel*) on top of Nse5/6. Colour coding for conserved residues as in (D). An extended loop in this region contains a lysine residue (K148) that was found to cross-link to Smc5 and Smc6 ATPase heads (see Fig 3).

Source data are available online for this figure.

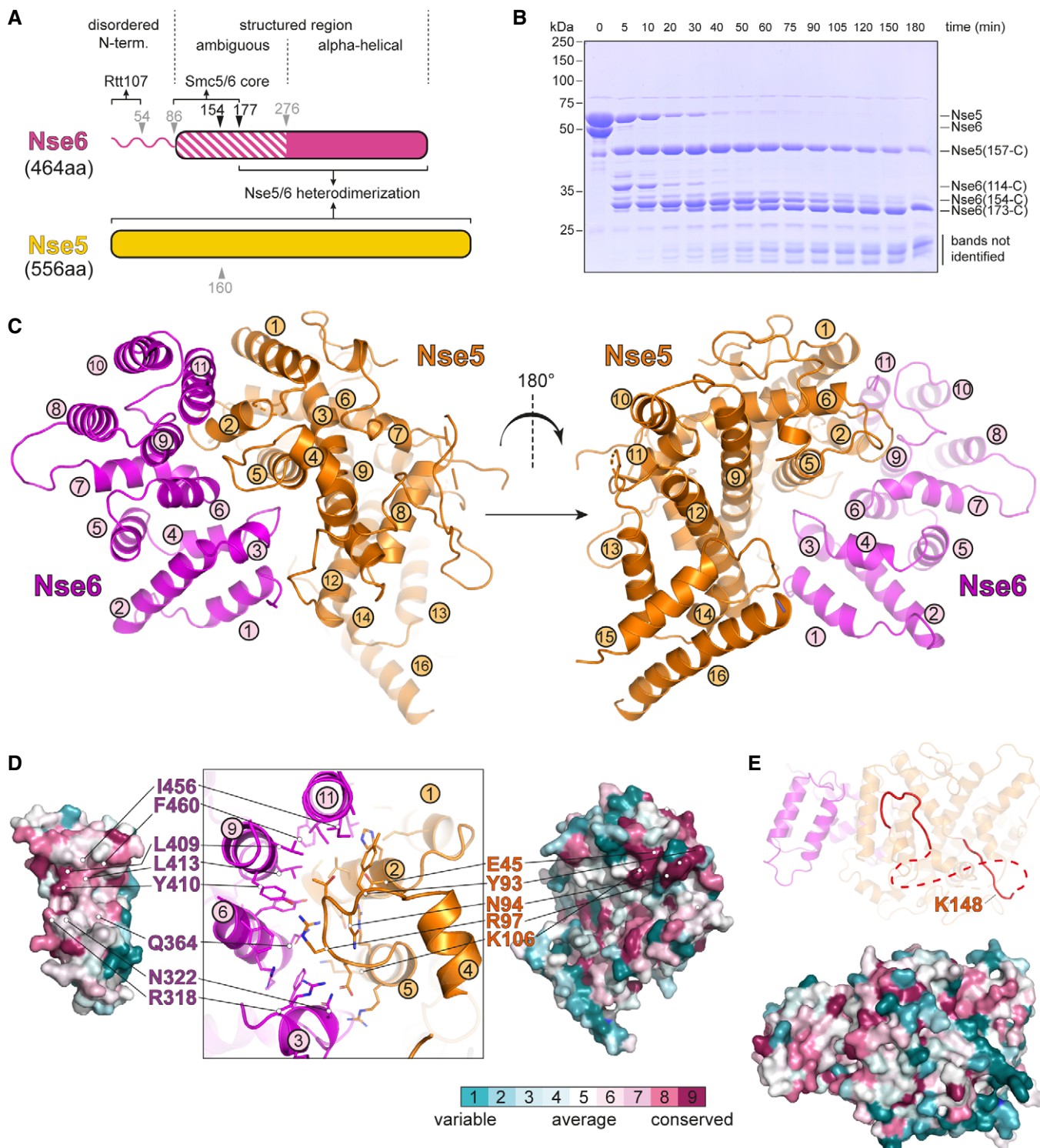

**Figure 2.**

Nse5, respectively. Moreover, we found that the binding of the Smc5/6 hexamer to Nse6 (1–179) was salt labile, while the binding to full-length Nse5/6 was resistant to washes with up to at least 600 mM NaCl. This indicates that additional sequences contribute to the formation of a stable octameric complex (Fig 3B). Supporting

this notion, we found that Nse5/6 outcompeted the N-terminal fragment of Nse6 in binding to the Smc5/6 hexamer (Fig EV2A) and was itself not dissociated from the hexamer by repeated washes with an excess of this Nse6 fragment (Fig EV2B). For Nse5, an N-terminal fragment (1–160) and the corresponding C-terminal

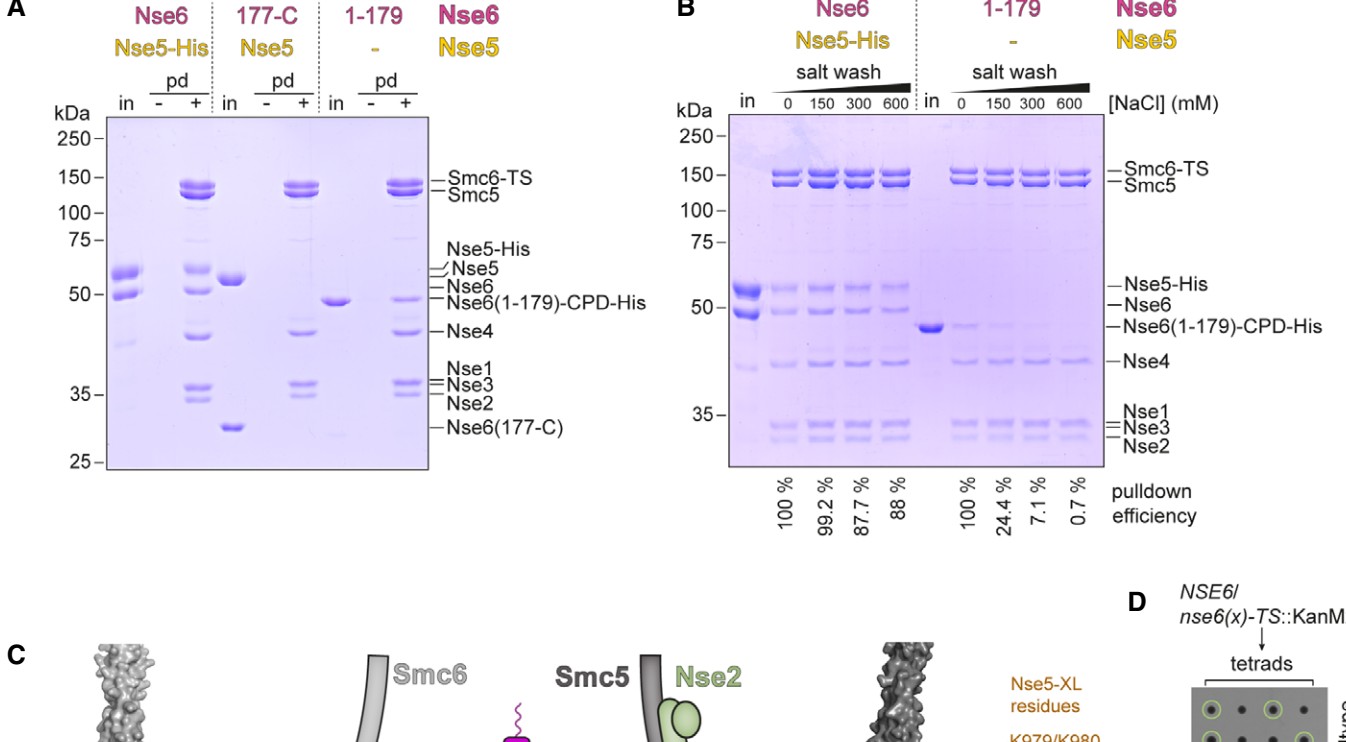

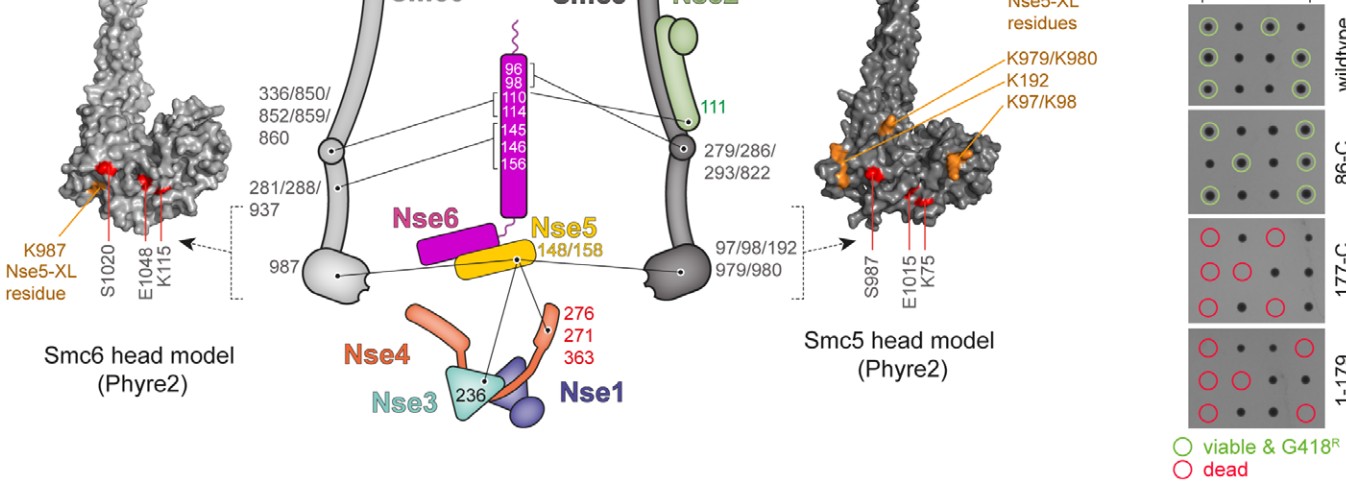

**Figure 3.  Association of the yeast Smc5/6 hexamer with the Nse5/6 dimer.**

A  Pulldown ("pd") assays using immobilized Smc5/Smc6-Twin-Strep ("TS") hexamers and soluble input material ("in") of Nse5/6 (*left*), of Nse5/Nse6(177-C) (*middle*) and of Nse6(1–179)-CPD-His (*right*). Control pulldowns ("−") were performed by omitting the pre-binding of Smc5/Smc6-Twin-Strep to the beads. Fractions were analysed by SDS–PAGE and Coomassie staining.

B  Salt stability of Smc5/Smc6-Twin-Strep interactions with Nse5/6 and Nse6(1–179). Immobilized Smc5/Smc6-Twin-Strep was mixed with Nse5/6 or Nse6(1–179)-CPD-His ("in"). Beads were washed with buffers containing the indicated salt concentrations. Bead fractions were analysed by SDS–PAGE and Coomassie staining. Pulldown efficiencies were estimated from the intensity of Coomassie gel bands.

C  XL-MS cross-links detected between Nse5 and Nse6 proteins and subunits of the Smc5/6 hexamer in schematic representation (middle panel). The positions of lysine residues on the Smc6 head and the Smc5 head, left and right panels, respectively, with cross-links to Nse5 are denoted on Phyre2-generated homology models.

D  Analysis of spore viability by yeast tetrad dissection. Diploid strains heterozygous for alleles of Nse6 (wt, 86-C, 177-C and 179-C) were sporulated. Isolated spores were grown on YPD plates. Viable clones were tested for the marker cassette conferring resistance to G418 (marked by circles in green colours). Dead spores were marked by circles in red colours.

Source data are available online for this figure.

fragment (160-C) turned out to be poorly expressed even when Nse6 was co-expressed, presumably due to the folding of Nse5 into a single globular domain as seen in the crystal structure (Fig 2C), thus preventing the mapping of binding sites by truncations.

In our XL-MS data, we detected multiple inter-domain cross-links between Nse5/6 and the Smc5/6 hexamer (Fig 3C), which are in line with the putative interface determined by pulldown assays and also with independent XL-MS experiments published while this

work was in progress (Gutierrez-Escribano *et al*, 2020; Yu *et al*, 2021). Of note, cross-links between Nse5 and Nse6 were surprisingly rare and only observed under the ATPase buffer conditions (see below). N-terminal sequences of Nse6 displayed several contact points with the hexamer, consistent with the binding of Nse6 (1–179) to Smc5/6 observed in our pulldown assays (Fig 3A). The observed inter-domain cross-links clustered on the Smc5 and the Smc6 joint regions as well as the joint-proximal part of the Nse2 protein. On the other hand, Nse5 residues cross-linked to the head domains of Smc5 and Smc6 as well as to the Nse1/3/4 sub-complex, suggesting that the Nse5/6 dimer bridges distal parts of the Smc5/6 hexamer by contacting the joint region via Nse6 sequences and the head region via Nse5 sequences. Intriguingly, several Nse5 inter-domain cross-links to Smc5/6 mapped to the direct vicinity of the Walker A and B box and signature motif residues. This suggests that Nse5/6 is located between the Smc5 and Smc6 heads, which is also supported by 3D-reconstructions from negative stain electron microscopy (Hallett *et al*, 2021). If so, it might interfere with the ATP hydrolysis cycle (see below). On the Nse5/6 structure, the two Nse5 lysine residues that form cross-links to the Smc5 and Smc6 head domains are located in a loop between helices α6 and α7. K148 is visible, while reliable electron density was not observed for K158 in our crystal. Interestingly, this loop protrudes from the structure next to a highly conserved surface patch (Fig 2E), supporting the hypothesis that this region is involved in interaction between Smc5/6 and Nse5/6 complexes.

To establish whether the Nse6 sequences mediating binding to Nse5 and Smc5/6 are required for cellular function, we next generated truncation mutants of the *nse6* gene in yeast by allelic replacement in a diploid strain. Isolation of haploid progenies by germination of spores showed that the Nse6(86-C) fragment was able to support apparently normal growth (Fig 3D), while Nse6 (1–179) and (177-C) failed to do so. Combined, the observations support that both the N-terminal Smc5/6 interacting region and the Nse5-binding C-terminus are crucial for correct function of Nse5/6 in yeast, while the extreme N-terminal sequences (1–84) known to bind to the DNA repair factor Rtt107 are dispensable (Wan *et al*, 2019). Curiously, we failed to detect any increased sensitivity of the *nse6(86-C)* mutant strain towards UV irradiation or treatment with

MMS or HU when assaying viability and growth by spotting (Fig EV2D). The Rtt107/Nse6N interaction is thus dispensable even when cells are challenged by elevated levels of DNA damage.

### Nse5/6 inhibits the Smc5/6 ATPase

We next characterized the ATP hydrolysis activity of purified preparations of Smc5/6 complexes. Under our experimental conditions (150 nM protein in ATPase buffer), the hexamer exhibited an ATP hydrolysis rate of approximately 40 ATP/min/complex in the absence of DNA, which is in the typical range for SMC complexes (Fig 4A). This basal activity was virtually unchanged by addition of plasmid DNA ("plasmidDNA"; 25 kbp; 250 bp DNA per hexamer) in closed circular form (a mixture of supercoiled and relaxed DNA; see Fig EV3E) or after linearization (Figs 4A and EV3D). As expected, an equivalent preparation harbouring a Walker B motif active site mutation ("EQ") in Smc5 and in Smc6 did not display noticeable ATP hydrolysis activity, confirming the absence of contaminating activities and underscoring the importance of the Walker B motif in Smc5 and Smc6 for ATP hydrolysis (Fig EV3A).

Addition of Nse5/6 strongly inhibited the Smc5/6 ATPase (Fig 4A). This inhibition was particularly strong at lower ATP concentrations and can thus largely be attributed to a reduction in the apparent affinity for ATP (i.e. an increased $K_m$ for ATP binding). At saturating ATP concentrations, the decrease in turnover ($k_{cat}$) was comparatively mild (approximately two-fold). This implies that Nse5/6 mainly inhibited the ATPase activity by precluding productive ATP binding by Smc5/6. It might do so either by interfering directly with the ATP binding step or by preventing subsequent head engagement, which completes the formation of the ATP binding pocket. Both scenarios are consistent with the proximity of Nse5 residues to the active site of Smc5 and Smc6 as detected by XL-MS (Fig 3C), but we favour the latter because it is in agreement with our cysteine cross-linking experiments (see below) and also supported by an independent study on the overall structure of the yeast Smc5/6 complex by negative stain electron microscopy (Hallett *et al*, 2021).

Unlike the hexamer, the Smc5/6 octamer showed robust stimulation of ATP hydrolysis by addition of plasmidDNA, yielding an ATP

▶

---

**Figure 4. ATP hydrolysis by purified Smc5/6 hexamer and octamer.**

A  ATP hydrolysis rates (given per Smc5/6 complex) were measured by an enzyme-coupled assay in the absence and presence of plasmidDNA (conc. 1.25 nM) for the Smc5/6 hexamer (conc. 150 nM) and the octamer (conc. 150 nM) with increasing concentrations of ATP. The curves were fitted to Michaelis–Menten equation, and $K_m$ and $k_{cat}$ values were determined. Please note that the hexamer without plasmidDNA shows cooperative behaviour; thus, the Michaelis–Menten kinetics is not formally applicable (marked by asterisk). Assays were performed in biological triplicates, and mean values are shown with error bars indicating standard deviations. For bar graphs, individual data points are also displayed.

B  Same as in (A) using 40bpDNA (annealed 40 bp oligo DNA) (conc. 1 μM) instead of plasmidDNA. Note that the data points and curves for samples without DNA (hexamer -DNA; octamer -DNA) are identical to (A).

C  Fluorescence anisotropy measurements using 40 bp dsDNA or 40-mer ssDNA substrates. Representative binding curves (top graph) and the resulting Kd values (bottom graph) indicate that both hexameric and octameric Smc5/6 complexes interact with both substrates, whereas the Nse5/6 complex alone does not. The bar graph shows mean values with standard deviations from technical triplicates. Individual data points are also displayed.

D  Pulldown experiments with Smc5/6 complexes and circular plasmid (2.8 kbp). DNA is retained after high salt washes (1 M NaCl) only in the presence of both Nse5/6 and ATP. The graph on the right shows a quantification (mean values and standard deviations) of the amount of co-purified DNA from technical triplicates. Individual data points are also displayed.

E  Salt-stable DNA association requires the DNA substrate to be circular ("circ"). Same experimental setup as in (D), but the plasmid substrate was also linearized by restriction digest ("lin").

F  Salt-stable DNA association requires ATP binding and Smc5/6 head engagement, but not ATP hydrolysis. Same experimental setup as in (D) using mutant versions of the Smc5/6 hexamer as well as the non-hydrolysable analogue of ATP, ATPγS, as indicated.

Source data are available online for this figure.

---

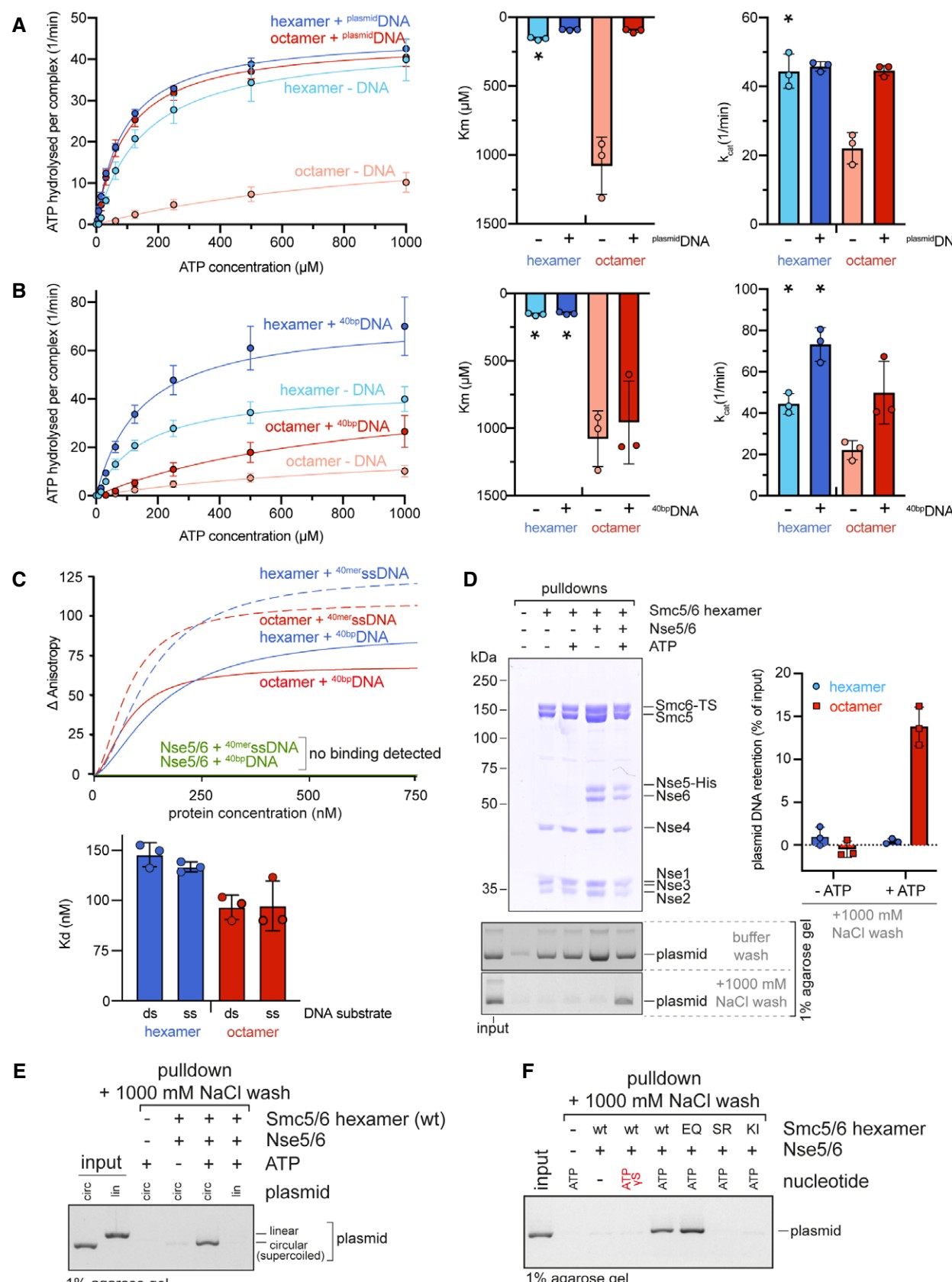

**Figure 4.**

hydrolysis rate of approximately 40/min/complex (Fig 4A). The presence of $^{plasmid}$DNA lowered the $K_m$ for ATP and increased the $k_{cat}$, essentially cancelling the inhibitory effects of Nse5/6 on the hexamer (Fig 4A). The addition of equivalent concentrations of 40 bp double stranded DNA ("$^{40bp}$DNA") (1 μM; i.e. 250 bp DNA per octamer) had little to no effect on $K_m$ (Fig 4B). Together, these observations indicate that only larger DNA molecules alleviate the inhibition of the Smc5/6 ATPase by Nse5/6, which suggests that a given DNA molecule might have to occupy multiple DNA binding sites simultaneously to efficiently stimulate ATP hydrolysis in the octamer. We hypothesize that DNA competes with Nse5/6 for binding to the Smc5/6 head domains thus counteracting Nse5/6 to allow for productive head engagement.

While $^{plasmid}$DNA stimulated the octamer more strongly when compared to $^{40bp}$DNA, the hexamer curiously showed the opposite response. Its $k_{cat}$ was mildly but clearly stimulated (roughly two-fold) by the addition of $^{40bp}$DNA (Figs 4B and EV3D). Intriguingly, the presence of $^{plasmid}$DNA hindered the stimulation of the hexamer by $^{40bp}$DNA (Fig EV3D), suggesting that both types of DNA molecule compete for same DNA binding interface, although only the less stably bound $^{40bp}$DNA leads to ATPase stimulation. The Smc5/6 hexamer furthermore exhibited cooperativity, i.e. the ATP hydrolysis rate per Smc5/6 complex increased with elevated protein concentrations (Fig EV3B). This implies that at least some of the observed ATP hydrolysis occurred in a dimer of hexamers or a higher oligomeric form, unlike for instance the ATP hydrolysis activity of B. subtilis Smc-ScpAB (Vazquez Nunez et al, 2019). The octamer did not show cooperativity in the measured range of concentrations (Fig EV3C), implying that Nse5/6 excluded productive association between Smc5/6 complexes. Also, the addition of $^{plasmid}$DNA eliminated cooperativity in the hexamer (Fig EV3C).

Smc5/6 has been reported to bind not only dsDNA but also ssDNA substrates (Roy et al, 2015). We thus tested the effect of 40-mer ssDNA on Smc5/6 ATPase activity. Whereas the hexamer responded mildly, no increased activity was detected for the octamer in the presence of this substrate (Fig EV3D). This is in agreement with a recently published study (Hallett et al, 2021). To rule out that this lack of effect was caused by a lack of interaction under our experimental conditions, we measured the affinity for dsDNA and ssDNA substrates by fluorescence anisotropy (Fig 4C). In agreement with independent reports (Hallett et al, 2021; Yu et al, 2021), the Nse5/6 dimer did not detectably bind DNA in the tested concentration range, whereas the Smc5/6 hexamer interacted strongly ($K_d$ around 150 nM) with both types of substrates. The reconstituted octamer also had no preference for either substrate and displayed a slightly higher affinity (lowered $K_d$ of around 100 nM), potentially because the Nse5/6 complex stabilizes a Smc5/6 conformation that is favourable for DNA binding.

## Nse5/6 enables salt-stable DNA association

Having confirmed that both hexameric and octameric Smc5/6 complexes have strong affinity for short DNA substrates, we wanted to also examine the binding mode of the complexes to more physiological substrates. A feature of many SMC complexes is their ability to bind to DNA in a salt-stable manner (Cuylen et al, 2011; Murayama & Uhlmann, 2014; Kanno et al, 2015; Collier et al, 2020), which in case of cohesin has clearly been attributed to topological

entrapment of DNA within the tripartite SMC/kleisin ring (Collier et al, 2020). ATP-dependent salt-stable binding was recently reported also for the purified Smc5/6 octamer from S. cerevisiae (Gutierrez-Escribano et al, 2020). To investigate a putative involvement of the Nse5/6 complex, we incubated the purified Smc5/6 complexes with a circular DNA substrate (a 2.8 kbp plasmid) with or without ATP, immobilized the proteins and analysed the co-isolation of DNA after washes with buffer containing either low or high salt concentration. The result clearly showed that while DNA was always co-purified after low salt washes, it was only retained during high salt washes when both Nse5/6 and ATP were present in the binding reaction (Fig 4D). The Nse5/6 complex thus is necessary for this type of DNA interaction, which furthermore requires the DNA substrate to be circular (Fig 4E) as reported previously (Gutierrez-Escribano et al, 2020). Future experiments using a covalently closed Smc5/Smc6/Nse4 rings will be needed to establish whether this binding involves entrapment in the SMC/kleisin ring.

Salt-stable plasmid binding was recently claimed to require ATP hydrolysis as it was not observed in the presence of the non-hydrolysable analogue ATPγS (Gutierrez-Escribano et al, 2020). We confirmed that ATPγS does not support salt-stable DNA binding (Fig 4F). Yet, we also found that a hydrolysis-deficient complex carrying Smc5(EQ) and Smc6(EQ) ("EQ") displayed efficient salt-stable DNA binding in the presence of ATP (Fig 4F). No DNA retention after high salt washes was observed with complexes carrying mutations preventing ATP binding ("KI") or head engagement ("SR"). Of note, we found by pulldowns that ATP but not ATPγS supports robust co-isolation of hinge-less Smc5(EQ) and Smc6(EQ) sub-complexes (Fig EV3F). ATPγS is thus likely poorly suitable to induce head engagement in yeast Smc5/6 (see also below). Our results thus demonstrate that salt-stable DNA interaction requires Nse5/6 as well as ATP-head engagement while ATP hydrolysis appears dispensable.

## Nse5/6 couples head engagement to DNA substrate selection

To test whether Nse5/6 modulates the Smc5/6 ATPase by hindering head engagement, we made use of site-directed chemical cross-linking with the thiol-specific short-spacer compound BMOE. All eight subunits of the Smc5/6 holo-complex naturally harbour cysteine residues (64 in total). Chemical cross-linking of these residues produced only subtle effects observed by SDS–PAGE (Fig 5A). Presumably due to intra-molecular cross-linking or mono-link modifications (i.e. one reactive group of the reagent reacted with a cysteine while the other group quenched), some of Nse4 migrated slightly slower through the gel (Fig 5A). At a low abundance, other cross-linked products appeared, particularly in the presence of $^{plasmid}$DNA, which likely leads to increased local concentration, and upon addition of Nse5 and Nse6. Otherwise, the electrophoretic mobility of proteins remained virtually unchanged indicating that the cross-linking of engineered cysteines might be readily detectable (Davidson et al, 2019).

As a proof-of-principle, we first engineered a pair of cysteine residues at the south-interface between Smc5 and Smc6 hinge domains (Hinge-Cys "south") based on structural information available for the fission yeast hinge (Alt et al, 2017) (Fig EV4A; PDB: 5MG8). BMOE treatment of purified Hinge-Cys preparations depleted Smc5 and Smc6 and generated a species that migrated slowly in

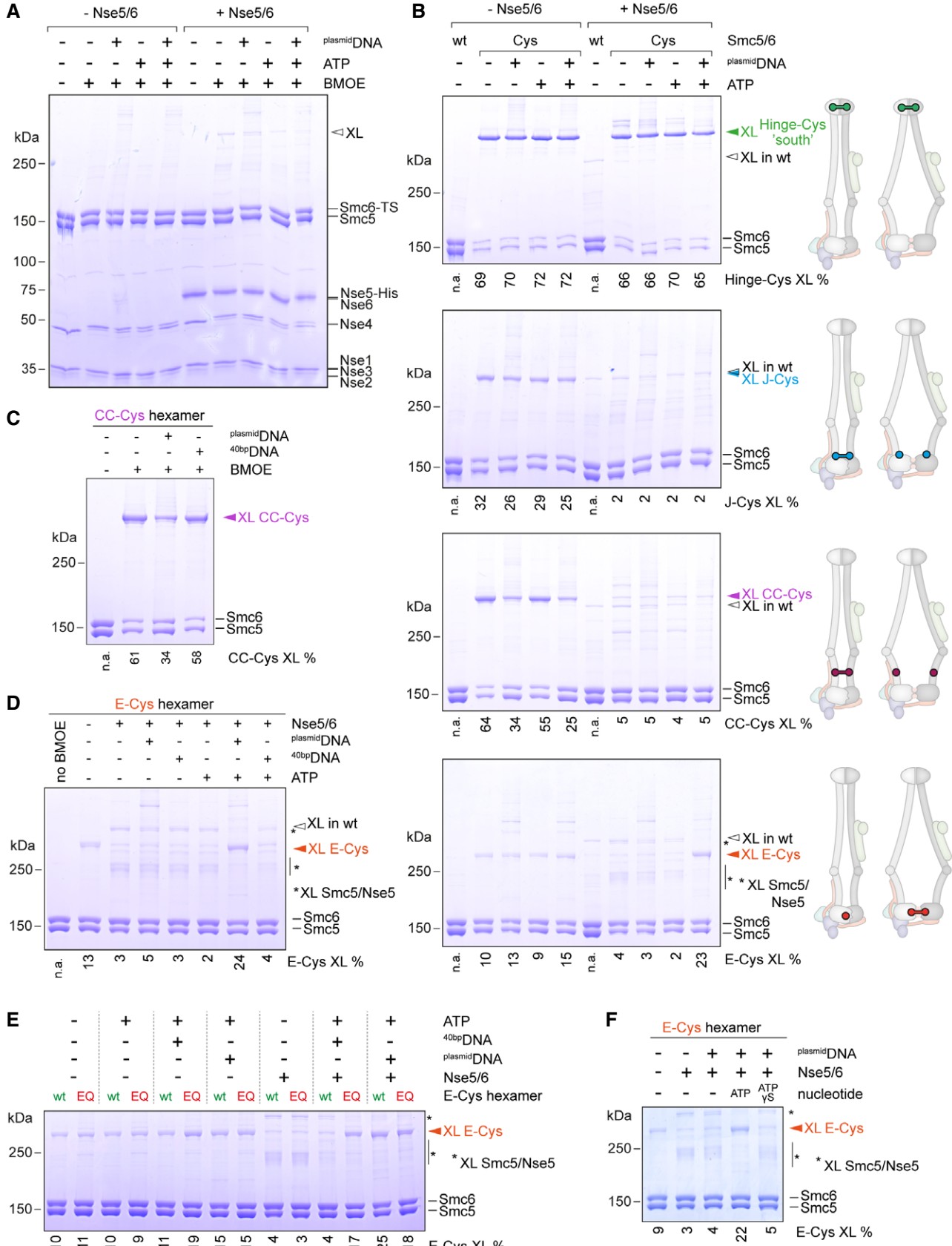

Figure 5.

Figure 5. Detection of Smc5/6 conformations by cysteine cross-linking.

A Cross-linking of a purified preparation of wild-type Smc5/6 hexamer in the absence and presence of Nse5/6. ATP and [plasmid]DNA were added as substrates as denoted. Products were analysed by SDS–PAGE and Coomassie staining. A cross-linked species derived from natural cysteines is indicated by an arrowhead ("XL"). Note that a fraction of Nse4 is converted into a more slowly migrating species, presumably by intra-molecular cross-linking.

B Cross-linking of engineered variants of the Smc5/6 hexamer. As in (A) using Smc5 and Smc6 cysteine mutants. Schemes indicate the location of engineered cysteines and their expected ability to cross-link in a rod-like and a ring-like conformation. High-molecular weight species were analysed by SDS–PAGE and Coomassie staining. Wild-type hexamer ("wt") is included as cross-linking control. Species occurring only in the presence of engineered cysteines are labelled by coloured arrowheads. Cross-linking efficiencies were calculated from the intensity of Coomassie-stained bands by comparing the band of the corresponding cross-linked species to the bands of unmodified Smc5 and Smc6. Numbers below the gel quantify the percentage of cross-linked protein species in the displayed gel. Comparable numbers were obtained in at least one additional independent experiment.

C DNA effects on Smc5/6 arm co-alignment as judged by CC-Cys cross-linking. As in (B) but also including [40bp]DNA. Addition of [plasmid]DNA but not [40bp]DNA reduces CC-Cys cross-linking in the Smc5/6 hexamer.

D DNA effects on head engagement as judged by E-Cys cross-linking. As in (B) also including [40bp]DNA. [plasmid]DNA binding, but not [40bp]DNA binding, overcomes inhibition of E-Cys cross-linking by Nse5/6.

E Effects of [40bp]DNA and [plasmid]DNA on head engagement (as judged by E-Cys cross-linking) of the Smc5/6 complex with wild type (wt) or hydrolysis-deficient (EQ) heads, in both the presence or the absence of the Nse5/6 complex. Cross-linking efficiencies were calculated from the intensity of Coomassie-stained bands by comparing the band of the corresponding cross-linked species to the bands of unmodified Smc5 and Smc6. Numbers below the gel quantify the percentage of cross-linked protein species in the displayed gel.

F Effects of [plasmid]DNA on head engagement of the Smc5/6 complex (as judged by E-Cys cross-linking) in the presence of ATP or the non-hydrolysable analogue ATPγS. Calculations and quantifications as in (E).

Source data are available online for this figure.

SDS–PAGE, corresponding to the molecular weight of the cross-linked Smc5-Smc6 product (Fig 5B). Hinge-Cys cross-linking was robust and unaffected by addition of ATP, [plasmid]DNA, or Nse5/6. A similar result, albeit with somewhat lower cross-linking efficiency, was obtained when cysteines were introduced at the "north" interface (Hinge-Cys "north"; Fig EV4A and B). Next, we designed a pair of cysteines to detect the juxtapositioned Smc5/6 heads (J-Cys) based on models of the rod-shaped conformation of the prokaryotic Smc-ScpAB complex (Diebold-Durand et al, 2017) (Fig EV4C and D). J-Cys cross-linking was clearly detected in the hexamer (Fig 5B). The cross-linking efficiency was not markedly altered by the presence of [plasmid]DNA or ATP. Addition of Nse5/6 however completely abolished J-Cys cross-linking regardless of the presence or absence of [plasmid]DNA and ATP. Loss of cross-linking upon addition of Nse5/6 was also observed with a pair of cysteines located in the head-proximal coiled coils (Figs 5B and EV4C). These cysteines (CC-Cys) were again designed based on published information for the prokaryotic Smc-ScpAB complex with the aim to detect co-aligned Smc5/6 arms. The cross-linking efficiency with CC-Cys was also significantly lowered (but not abolished) by the addition of [plasmid]DNA (Fig 5B) but not [40bp]DNA (Fig 5C) or ATP. These results confirmed our hypothesis that Nse5/6 alters the organization of Smc5/6 heads as well as the head-proximal coiled coils. Binding of Nse5/6 appears to destabilize the rod conformation (J-state) as judged by J-Cys and CC-Cys cross-linking.

As Nse5/6 interferes with ATP hydrolysis of Smc5/6, we next engineered cysteines for the detection of head engagement by BMOE cross-linking. The first attempted design based on an equivalent reporter in bacterial Smc-ScpAB failed due to instability of the mutant proteins (Minnen et al, 2016). Thus, we utilized naturally occurring cysteines on the Smc6 head (C92 and C147) and introduced a complementary cysteine residue on Smc5 (N975C), so that a pair of cysteines (E-Cys) would get cross-linked upon ATP-engagement (Fig EV4D). E-Cys cross-linking was detected at low levels in the hexamer with limited influence by ATP and by [plasmid]DNA (Fig 5B). This suggest that the Smc5 and Smc6 heads are arranged in a relatively flexible manner in the hexamer and are able to adopt both the J-state and the E-state. Like J-Cys and CC-Cys

cross-linking, also E-Cys cross-linking was strongly reduced by the addition of Nse5/6 (Fig 5B). Of note, minor levels of other cross-linked species appeared that were putatively derived from off-target cross-linking between residue N975C in Smc5 and an endogenous cysteine on Nse5. More importantly and unlike J-Cys and CC-Cys cross-linking, E-Cys cross-linking was boosted when ATP and [plasmid]DNA were supplemented in the presence of Nse5/6 (Fig 5B). This suggests that Nse5/6 hinders head engagement in the absence of a suitable DNA substrate and promotes engagement in its presence. [40bp]DNA was not able to substitute for [plasmid]DNA in overcoming the inhibition by Nse5/6 (Fig 5D) as expected from its failure to stimulate ATP hydrolysis by the octamer efficiently (Fig 4B).

We conclude from these results that binding of Nse5/6 to the Smc5/6 hexamer reorganizes the Smc5/6 head module in a manner that reduces the occupancy of the J-state and of the E-state by stabilizing yet another conformation. As Nse5/6 may keep Smc5 and Smc6 heads apart by intercalating between them, possibly analogous to the apo-bridged conformation of the Smc2 and Smc4 heads observed with yeast condensin (Lee et al, 2020), we tentatively call this conformation the "inhibited" conformation (with the heads occupying the "I-state"). Evidence for the intercalation of Nse5/6 between the heads was recently reported based on lower resolution 3D-reconstruction from negative stain electron microscopy (Hallett et al, 2021). Only in the presence of ATP and [plasmid]DNA, the inhibitory effect of Nse5/6 is overcome making head engagement favourable.

Occupancy of the E-state is expected to be increased in the case of the ATP hydrolysis-deficient Smc5(EQ)/Smc6(EQ) complex. However, we failed to detect a noticeable increase E-Cys cross-linking when introducing the EQ mutations (Fig EV4E). Addition of DNA substrates, however, revealed interesting differences. While the wild-type version did not respond to short DNA, its EQ counterpart clearly showed increased E-Cys cross-linking with [40bp]DNA in the context of the hexamer and the octamer (Fig 5E). Notably, no effect on E-Cys cross-linking was observed when adding ATPγS, in contrast to ATP (Fig 5F), providing additional support for the inability of this modified nucleotide to promote efficient head engagement.

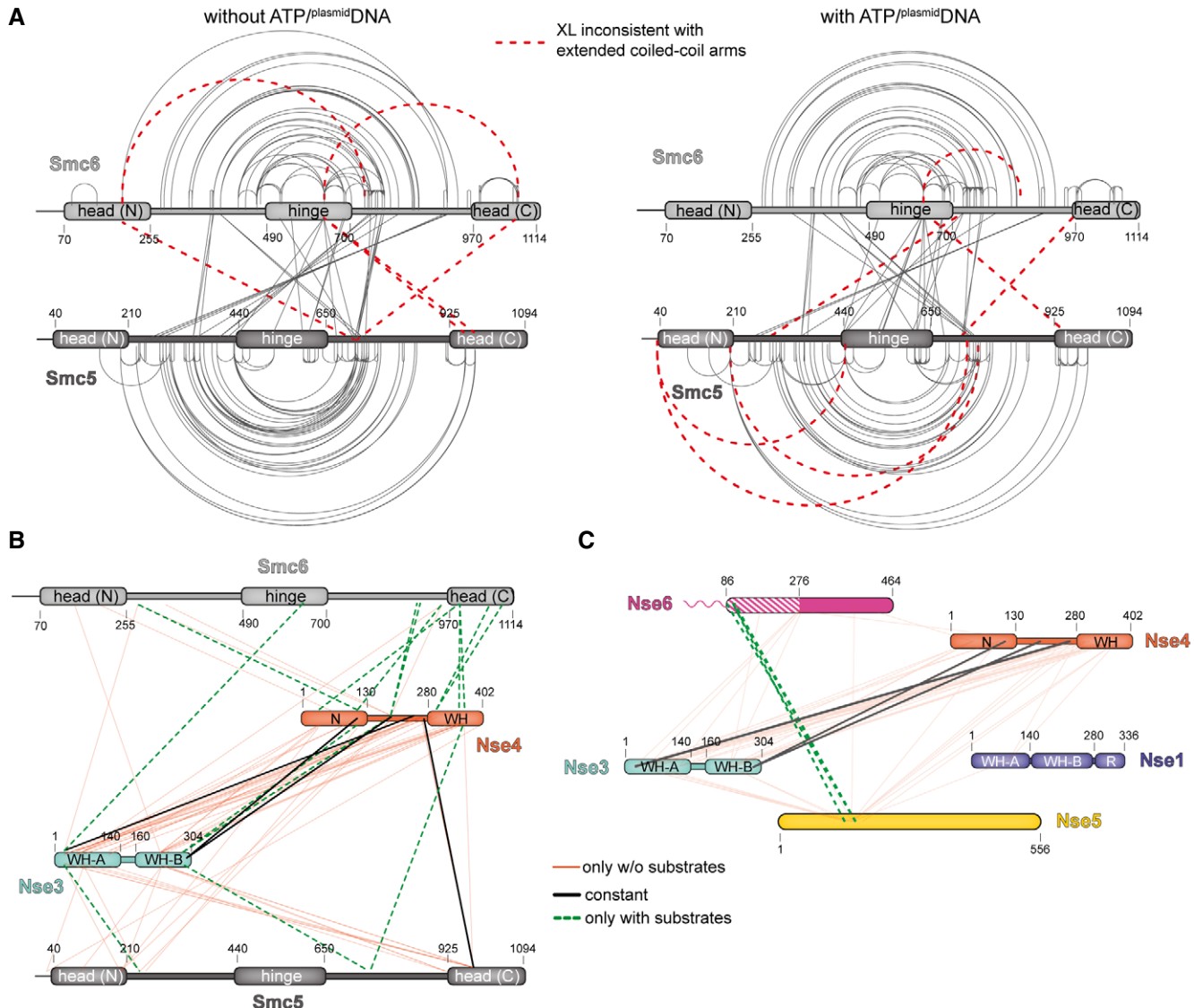

**Figure 6.  Alterations in the Smc5/6 architecture upon binding of ATP and ^plasmidDNA.**

A   Cross-links in the Smc5/6 octamer identified by XL-MS without substrate addition (*left panel*) and with ^plasmidDNA and ATP (*right panel*). As in Fig 1C using ATPase buffer. Only intra- and inter-links of the Smc5/6 dimer are displayed here. Other cross-links are shown in panel (B), (C), and Fig EV5 and listed in Dataset EV1.

B   Cross-links of Nse3/4 detected by XL-MS. Same experiment as in (A) using the Smc5/6 octamer in ATPase buffer. Cross-links between Nse3 and Nse4 as well as their cross-links to Smc5 and Smc6 are displayed. Lines in black colours indicate cross-links observed with and without substrates, in oranges colours only without substrates and in dashed lines in green colours only with substrate addition. See Fig EV5B for an alternative representation in circular plots.

C   Cross-links of Nse1/3/4 to Nse5/6. Same experiment as in (A) and (B). Display as in (B). See Fig EV5B for an alternative representation in circular plots.

## Global changes in Smc5/6 architecture upon ATP and ^plasmidDNA binding

To detect conformational changes in an unbiased manner, we next performed XL-MS on Smc5/6 octamers in the absence and presence of ATP and ^plasmidDNA. For these experiments, we switched to using the ATPase buffer (10 mM Hepes-KOH pH 7.5; 150 mM K-OAc; 2 mM MgCl$_2$, 20% glycerol) so that the conditions matched those used for the ATP hydrolysis measurements. In general, we observed more cross-links (207 intra-links and 169 inter-links) under these conditions than with the buffer used for our initial XL-MS experiments (20 mM Hepes-KOH pH 7.5; 250 mM NaCl). The cross-links in the absence of substrate were however largely comparable with the previously obtained XL-MS data (Fig 1B). Some additional cross-links were detected that appeared to deviate from the Smc5/6 rod pattern, many of which exhibited however relatively low MS cross-link identification scores (Fig 6A). Moreover, their positions were difficult to reconcile with one another and with the position of an elbow inferred from predicted discontinuities in the heptad register (Burmann *et al*, 2019). We thus presume that these cross-links

were (at least largely) derived from spurious contacts between hexamers, which might be favourable in the ATPase buffer, or arise from the pool of 1% false positives.

The total number of intra-links as well as the identity of most cross-links were also comparable in the presence and absence of the substrates. Smc5/6 inter- and intra-domain cross-links were not obviously changed, implying that any substrate-induced structural changes during the ATP hydrolysis cycle (such as head engagement and putative arm opening) were not captured by the lysine-specific cross-linking reagents (Fig 6A). Nse2 forms inter-domain cross-links to both Smc5 and Smc6, which were also virtually unchanged by addition of the ligands (Fig EV5A). However, we observed striking differences in the pattern for other modules. Most importantly, a majority of Smc5 and Smc6 inter-domain cross-links to Nse3 and to Nse4 were lost upon addition of ATP and [plasmid]DNA. They were replaced by a low number of new contacts (Fig 6B). Inter-domain cross-links between Nse3 and Nse4 were also dramatically reduced under these conditions. Similar and even more pronounced differences were observed for interactions between the Nse1/3/4 and Nse5/6 modules, which were well connected in the absence of ligands but not in their presence (Fig 6C). Despite the dramatic loss of cross-links between Nse1/3/4 modules with other components of the complex, it was still stably bound in pulldown experiments under these experimental conditions (Fig EV5E). Whereas some differences were also observed for cross-links between Nse5/6 and the Smc5 and Smc6 proteins, they were less pronounced and the interaction between the Nse6 N-terminus and the Smc5/6 joint region appeared unaffected (Fig EV5B). These results suggest that the Nse1/3/4 module is re-arranged and released from both the Nse5/6 module and the Smc5/6 heads in response to head engagement and [plasmid]DNA binding. Very similar results were obtained when both Smc5 and Smc6 heads carried the "EQ" mutation, demonstrating that ATP hydrolysis is not required for these global structural changes (Fig EV5C and D).

# Discussion

In this work, we uncovered a strong impact Nse5/6 has on the organization of the Smc5/6 complex and on the regulation of ATP hydrolysis, the essential enzymatic activity of the complex.

Our data show that Nse5/6 has multiple contact points with the Smc5/6 complex. The main anchor appears to be formed by physical association of the N-terminus of Nse6 with the Smc5/6 dimer as shown by pulldowns (Fig 3A and B) and supported by recently published and our new XL-MS data (Figs 3C and 6B) (Gutierrez-Escribano et al, 2020; Yu et al, 2021). The latter imply an interaction of the Nse6 N-terminus with the Smc5/6 joints, which is consistent with a domain mapping study using fission yeast proteins (Palecek et al, 2006) but appears to contradict a mapping study based on budding yeast proteins which suggested that Nse5/6 may associate with the Smc5/6 hinge (Duan et al, 2009b). Binding assays under high salt conditions and with competition furthermore indicated that additional Nse5 or Nse6 sequences strengthen the association in the octameric complex (Figs 3B and EV2). In the XL-MS experiments, such contact points were mapped, for example near the active sites of Smc5 and Smc6 and at a Nse5 surface located opposite of the Nse6 N-terminus (Fig 3C) (Gutierrez-Escribano et al, 2020). An

equivalent interaction has recently also been mapped between human Smc5/6 and Slf2 (Nse6) by Yeast-Two-Hybrid assays (Adamus et al, 2020), while there is no evidence for Smc5/6 binding by Slf1 (Nse5). Whether the yeast and human interfaces indeed share a similar fold and a common origin remains to be determined due to the absence of recognizable sequence conservation in the relevant regions.

We found that Nse5/6 strongly inhibits the basal activity of Smc5/6, either by directly blocking binding of ATP through interactions with active sites on Smc5 and Smc6 or by sterically blocking ATP-engagement of the heads. Both scenarios would hinder productive ATP binding explaining the poor ATP hydrolysis at reduced concentrations of ATP (Fig 4A). We provide several lines of evidence suggesting that the inhibition of the Smc5/6 ATPase is brought about by major conformational changes. As judged by cysteine cross-linking data, association of Smc5/6 with Nse5/6 eliminated the J-state regardless of substrate availability. It also abolished the E-state except in the presence of both ATP and [plasmid]DNA. We propose that this inhibited conformation (the I-state) might resemble the apo-bridged conformation of condensin where a hawk subunit (Ycs4) distantly bridges the Smc2 and Smc4 heads (Lee et al, 2020). The notion that productive head engagement is inhibited by Nse5/6 is in agreement with the positioning of Nse5/6 between the Smc5/6 head domains as indicated by recent electron microscopy data (Hallett et al, 2021). Since the Nse5/6 structure revealed rudimentary structural similarities with hawk proteins, it remains a (remote) possibility that these proteins share a common origin and similar functions.

The SUMO-ligase activity of Nse2 in an isolated Smc5/Nse2 sub-complex has intriguingly been reported to be controlled by DNA binding, with the Smc5 arms reorganizing upon DNA binding concomitant with Nse2 stimulation (Varejao et al, 2018). And Nse5 is known to interact with the SUMO protein, the substrate of Nse2, via SIM sequences. It might thus contribute to the SUMOylation of target proteins (Bustard et al, 2016; Yu et al, 2021). To reconcile these published and our new observations, we propose that the Nse5/6 complex, while being fastened onto the Smc5/6 joints by the Nse6 N-terminus, can reach out—putatively via a flexible connection in Nse6—either to inhibit the Smc5/6 ATPase located on one side of the Smc5/6 joints or to stimulate the Nse2 SUMO-ligase located on the opposite side by binding to the substrate SUMO (Fig 7). DNA binding would consequently activate not only the Smc5/6 ATPase by releasing the Nse5/6 inhibition but also stimulate target SUMOylation by Nse2. If so, then Nse5/6 would be a key regulator of at least two (of the three known) enzymatic activities in Smc5/6.

The Nse1/3/4 module—including the Nse1 ubiquitin ligase, the third enzymatic activity of Smc5/6—is also directly affected by [plasmid]DNA binding as judged by our XL-MS data (Fig 6). Multiple inter-domain cross-links of the Nse1/3/4 module to the Smc5 and Smc6 proteins, mostly to the heads, were lost upon ATP and [plasmid]DNA addition. We postulate that Nse1/3/4 relocates by binding to DNA using the proposed DNA-binding residues in Nse3 (Zabrady et al, 2016; Vondrova et al, 2020), possibly assisting to evict Nse5/6 from the heads and if so underscoring the central role of Nse5/6 in regulating the enzymatic activities of the Smc5/6 complex.

Smc5/6 hexamer and octamer showed disparate basal ATP hydrolysis activity and responded distinctly to addition of [40bp]DNA

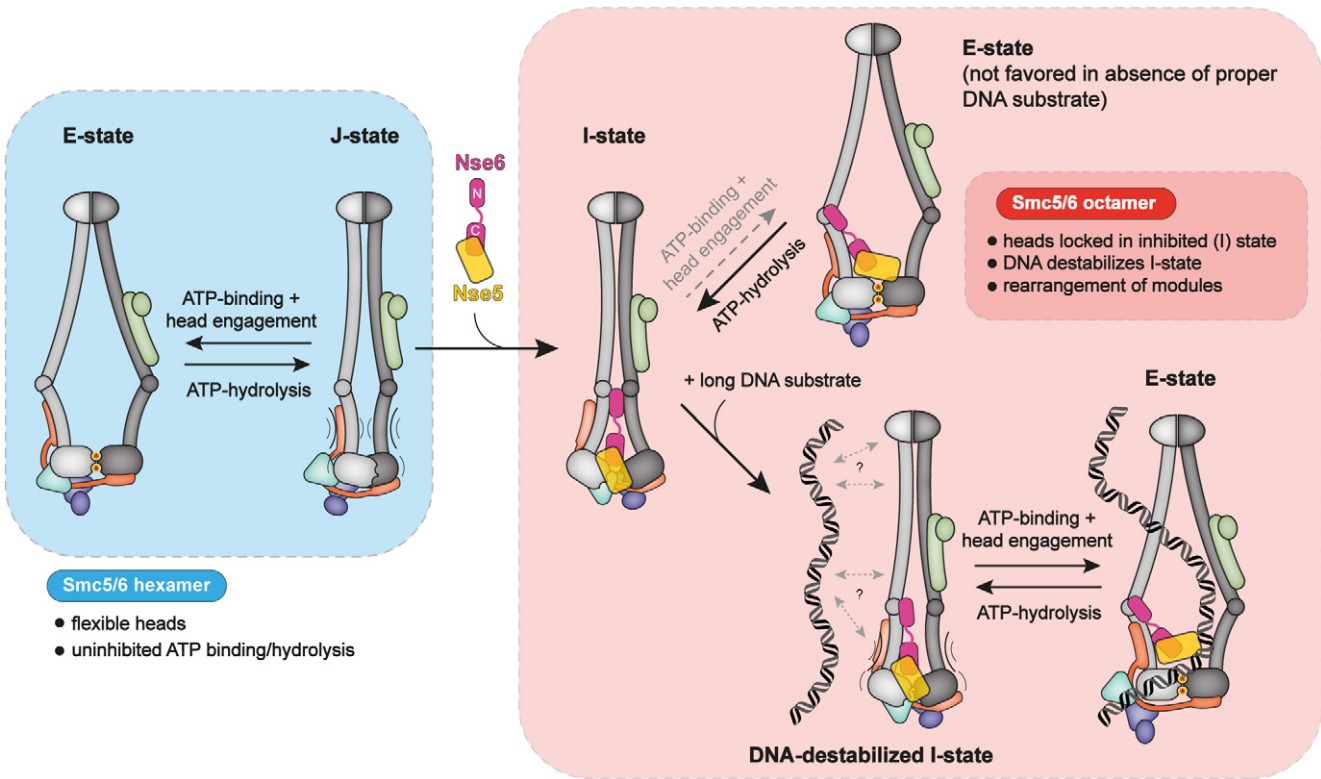

**Figure 7.  Model for conformational changes in Smc5/6..**

and [plasmid]DNA. Nse5/6 hindered short linear DNA molecules from activating the Smc5/6 ATPase, presumably by blocking access to a DNA binding site, possibly the head/DNA binding interface. Curiously, like Nse5/6, also [plasmid]DNA molecules blocked [40bp]DNA from stimulating ATP hydrolysis. These findings suggest that multiple mechanisms of DNA stimulation exist with one predominating in the hexamer, the other in the octamer. Moreover, only the hexamer exhibited cooperativity in ATP hydrolysis. Clearly, Nse5/6 has a major impact on Smc5/6 activity. We thus suspect that Nse5/6 (or Slf1/2) is needed for proper activity of this complex. It will be interesting to compare hexamer and octamer activity in recently reported single-molecule assays (Gutierrez-Escribano *et al*, 2020; Serrano *et al*, 2020). Inhibition of a SMC ATPase by an associated subunit has previously been reported for *E. coli* MukBEF (Zawadzka *et al*, 2018). Like with Nse5/6, the inhibition by MukE is counteracted by DNA binding. Whether MukE and Nse5/6 have similar roles in DNA substrate selection and similar mechanisms, remains to be determined.

Our DNA association studies show that Nse5/6 and ATP-dependent head engagement are strictly required for salt-stable interaction of the Smc5/6 complex with a circular DNA substrate, potentially by topologically trapping the DNA within a ring-shaped protein compartment. In contrast to a previous study (Gutierrez-Escribano *et al*, 2020), however, we find that ATP hydrolysis is dispensable for the DNA interaction, as previously observed for the cohesin DNA gripping state (Camdere *et al*, 2018; Collier *et al*, 2020; Higashi *et al*, 2020). In stark contrast to the observed inhibition of the Smc5/6 ATPase by Nse5/6, the cohesin loader, Scc2/4 (NIPBL), activates the

cohesin ATPase (Petela *et al*, 2018; Davidson *et al*, 2019). If Nse5/6 and Scc2/4 are indeed bona fide loaders for their respective complexes, then their loading mechanisms might be starkly different, apart from the fact that both loader complexes directly influence these SMC complexes at their cores, the ATPase heads.

The arms of cohesin, condensin and the non-canonical bacterial SMC complex MukBEF have been unequivocally shown to fold ~180° at an elbow (Burmann *et al*, 2019; Lee *et al*, 2020). The position of the elbow along the arms and its sequence are only poorly conserved across these divergent families of SMC complexes, implying independent emergence. The possible existence of elbows in the arms of yeast Smc5/6 was previously reported, but not without controversy. Here, we found no convincing evidence for the folding of arms in yeast Smc5/6, with similar conclusions recently being drawn for the same complex (Yu *et al*, 2021) and a human Smc5/6 core complex (Serrano *et al*, 2020). While some XL-MS inter-domain cross-links are consistent with a folded state, these cross-links imply different folding points, more often than not being inconsistent with the predicted discontinuities in the heptad repeat of the Smc5/6 coiled coils (Burmann *et al*, 2019), presumed to support the arm folding (found one third of the way from the hinge). Together with the display of fully extended particles by electron microscopy, we argue that the unfolded state is predominating in our preparations of complexes and the presence of the inter-domain cross-links rather point to inter-complex interactions or false positives which cannot be excluded due to the false positive rate correction at 1% employed in the data analysis. Similarly, no obvious indication has been found for a folded state of the ubiquitous bacterial SMC complex

Smc-ScpAB (Soh *et al*, 2015). Curiously, ancestral forms of Smc-ScpAB and of Smc5/6 are thought to be at the root of prokaryotic and eukaryotic SMC complexes, respectively, again suggesting that elbows have emerged independently at least twice (in ancestral MukBEF and in a common ancestor of cohesin/condensin) (Palecek & Gruber, 2015; Wells *et al*, 2017). If these considerations are valid, then the elbow likely provides for optimized SMC activity rather than being required for its core function.

Altogether, we have uncovered a central role of the Nse5/6 "loader" complex in the regulation of the Smc5/6 ATPase and DNA-binding activities. With Nse5/6 being specific to the family of Smc5/6 complexes only, we believe the new insights are directly relevant in specifying the unique activities of this complex in genome maintenance and chromosome segregation.

# Materials and Methods

### Preparation of DNA substrates

Two types of DNA substrates were used in this study. A 25 kbp plasmid ($^{plasmid}$DNA; pSG4050) was purified using a NucleoBond Xtra Maxi EF kit (Macherey Nagel) according to the manufacturer's instructions. The plasmid stock had a concentration of 1.5 mg/ml, equivalent to around 90 nM plasmid. Linearized plasmid was prepared by incubation of the circular substrate with AgeI restriction enzyme, and the enzyme was afterwards inactivated by heating to 80°C for 20 min. The second type of DNA substrates was a 40 bp duplex ($^{40bp}$DNA) obtained by annealing of two complementary oligonucleotides (STI699: 5′- TTAGTTGTTCGTAGTGCTCGTCTGG CTCTGGATTACCCGC-3′, STI700: 5′- GCGGGTAATCCAGAGCCAGA CGAGCACTACGAACAACTAA-3′). The $^{40bp}$DNA stock solution was prepared at a concentration of 100 μM.

### Yeast strains

All yeast strains were in the W303 background and were created using standard methods.

| | |
|---|---|
| YSG0008 | MATa/a, ade2-1, trp1-1, can1-100, leu2-3,112, his3-11,15, ura3, GAL, psi+ |
| YSG0232 | as YSG0008, but with NSE6/nse6Δ::KanMx |
| YSG0257 | as YSG0008, but with NSE6/NSE6-3CTSHis::KanMx |
| YSG0259 | as YSG0008, but with NSE6/nse6(86-454)-3CTSHis::KanMx |
| YSG0260 | as YSG0008, but with NSE6/nse6(177-454)-3CTSHis::KanMx |
| YSG0261 | as YSG0008, but with NSE6/nse6(1-179)-3CTSHis::KanMx |

### Protein expression in *E. coli*

All proteins and protein complexes described in this manuscript were expressed in *E. coli* (DE3) Rosetta. Hexameric Smc5/6 complexes were produced by protein expression from a single plasmid (based on the pET series of vectors) carrying all six subunits, with a C-terminal 3C-Twin-Strep-tag on Smc6. The four Nse subunits (Nse1, Nse2, Nse3 and Nse4) were sequentially fused with sequences containing ribosomal binding sites using standard techniques, thus creating a tetracistronic module (Nse1-RBS-Nse3-RBS-Nse4-RBS-Nse2). To generate the final plasmid, we first created individual gene expression cassettes (GECs) smaller parts (promoter, terminator, coding sequence, tag) using standard Golden-Gate assembly. Smc5 transcription was regulated with a tac-promoter and lambda-terminator, while the other subunits were transcribed by T7 promoters and terminators. These GECs were subsequently inserted into the final vector by Gibson-Assembly using appropriate terminal regions of homology. The final order of GECs in the resulting vector was (i) Smc6-3C-TS, (ii) Smc5 and (iii) the tetracistronic Nse1-4 construct, and the vector was based on a pET-28 backbone with a kanamycin-resistance cassette. For expression plasmids of mutant versions, we exchanged the coding sequence of the wild type with one containing the desired mutation prior to expression vector assembly.

For production of Nse5/6 complex, several tagged versions were prepared. In all cases, the two coding sequences were connected by a ribosomal binding site to create a bicistronic construct (Nse6-RBS-Nse5) driven by a T7 promoter. Affinity tag(s) were added to the N-terminus of Nse6 and/or the C-terminus of Nse5. For pulldowns between Twin-Strep-tagged Smc5/6 hexamer and Nse5/6 dimer, a version containing a C-terminal 3C-His(8) tag on Nse5 was used. For ATPase assays, Nse6 carried an N-terminal Twin-Strep-tag, and Nse5 contained a C-terminal His(8) tag. Reconstitution of octameric complexes was performed with a C-terminally AviTag-3C-Twin-Strep-tagged version of Nse5. Crystallization constructs with N-terminally truncated versions of Nse6 carried an N-terminal His (10)-Twin-Strep-3C-tag on Nse6. The N-terminal Nse6 fragment used for pulldown assays with the Smc5/6 hexamer was produced with a C-terminal Cysteine Protease Domain (CPD-His) tag.

For all purifications, 1 l of the strain carrying the desired plasmid was grown in TB-medium at 37°C to an OD(600 nm) of 1.0 and the culture temperature was reduced to 22°C. Expression was then induced with IPTG at a final concentration of 0.4 mM and allowed to proceed overnight (typically for 16 h).

### Purification of the hexameric *S. cerevisiae* Smc5/6 complex

*E. coli* cells were harvested by centrifugation and resuspended in 3–4 × the pellet volume of lysis buffer (50 mM Tris–HCl pH 7.5, 300 mM NaCl, 5% glycerol, 25 mM Imidazole) freshly supplemented with 5 mM DTT, 1 mM PMSF and 750 units of SM nuclease. All subsequent buffer except the gel filtration buffer contained 2 mM of DTT. Cells were lysed by sonication on ice with a VS70T tip using a SonoPuls unit (Bandelin) at 40% output for 15 min with pulsing (1 s on / 1 s off), typically yielding a total delivered energy of 15 kJ. The lysate was clarified by centrifugation (40,000 *g* for 30 min) and the supernatant applied onto a 5 ml StrepTrap column (GE Healthcare). After washing with 10 column volumes (CV) of lysis buffer the bound material was eluted with 4 CV of lysis buffer supplemented with 2.5 mM desthiobiotin and fractions of 1.5 ml were collected. Fractions containing the complex were then applied onto a 5 ml HiTrap Heparin column (GE Healthcare), and after washing with 5 CV of lysis buffer, the bound material was eluted with 4 CV of Heparin elution buffer (20 mM Tris pH 7.5, 1,000 mM NaCl, 2 mM DTT). Fractions of 1.5 ml were collected and those containing the target were concentrated with Amicon Ultra centrifugal filter units (50 kDa cutoff; Millipore) to a final concentration of

around 8 mg/ml (20–25 μM). The protein was then injected onto a Superose6 10/300 GL size-exclusion chromatography (SEC) column. The standard SEC buffer contained 20 mM Tris–HCl pH 7.5, 250 mM NaCl and 1 mM TCEP. For experiments involving lysine cross-linking, a buffer containing 20 mM Hepes-KOH pH 7.5 was used instead of Tris. Fractions containing the complex were concentrated to around 5 μM and snap-frozen in small (30–50 μl) aliquots for subsequent experiments.

**Purification of full-length Nse5/6 complexes**

Lysates were prepared as described for the Smc5/6 hexameric complex, and the same lysis buffer was used with the exception that 5 mM DTT was replaced by 5 mM beta-mercaptoethanol. For the version with a C-terminal 3C-His(8) tag on Nse5, the lysate was first loaded onto a 5 ml HisTrap column (GE Healthcare), and after washing with 5 CV of lysis buffer and 5 CV of buffer A (20 mM Tris–HCl pH 7.5, 100 mM NaCl, 5 mM beta-mercaptoethanol), the bound material was eluted with a 10 CV gradient from buffer A to the same buffer supplemented with 500 mM Imidazole pH 7.5. Fractions containing the complex at reasonable purity were loaded onto a 5 ml HiTrap Heparin column (GE Healthcare), and the bound protein was eluted with a gradient from buffer A to buffer B (20 mM Tris–HCl pH 7.5, 1,000 mM NaCl, 2 mM DTT). Fractions with protein at sufficient purity were concentrated and injected onto a Superdex200 Increase 10/300 GL SEC column in a buffer containing 20 mM Tris pH 7.5, 300 mM NaCl and 1 mM TCEP. The final protein was concentrated to around 20–30 μM and snap-frozen in small (20–30 μl) aliquots.

For the version that contained an N-terminal Twin-Strep-tag on Nse6 and a C-terminal His(8) tag on Nse5, the purification was performed as described for the Nse6/Nse5-3C-His complex, except that the elution from the HisTrap column was loaded onto a 5 ml StrepTrap column and bound material was eluted with buffer A supplemented with 2.5 mM desthiobiotin. All subsequent steps (Heparin and SEC) were identical.

For the version containing a C-terminal AviTag-3C-Twin-Strep-tag on Nse5, the lysate was loaded onto a 5 ml StrepTrap column and eluted with buffer A containing 2.5 mM desthiobiotin. The elution was then loaded on a Heparin and SEC column as described for the other complexes.

For crystallization constructs with N-terminally His(10)-Twin-Strep-3C-tags, the lysate the purification was identical to the Twin-Strep-Nse6/Nse5-His version, except that before SEC the tag was removed by incubation with HRV-3C protease (overnight at 4°C) and that a different SEC buffer (20 mM Tris pH 7.5, 500 mM NaCl, 1 mM TCEP) was used.

Nse6(1-179)-CPD-His protein was purified as described for the Nse6/Nse5-3C-His version.

**Reconstitution of octameric complexes by size-exclusion chromatography**

Purified Smc5/6 hexamer (with a C-terminal 3C-Twin-Strep-tag on Smc6) and Nse5/6 dimers (various versions) were mixed in a total volume of 500 μl with a 1.5× molar excess of Nse5/6. This mixture was then subjected to SEC using either a Superose6 Increase 10/300GL or a Superose6 Increase 3.2/300 column. For most experiments, the Nse6/Nse5-3C-His(8) complex was used, the tags were not removed, and the SEC buffer contained 20 mM Tris–HCl pH 7.5, 250 mM NaCl and 1 mM TCEP. For lysine cross-linking analysis, the hexamer was mixed with Nse6/Nse5-AviTag-3C-Twin-Strep, the Twin-Strep-tags were removed by incubation with HRV-3C protease overnight at 4°C before SEC, and an amine-free SEC buffer (10 mM Hepes-KOH pH 7.5, 250 mM NaCl, 1 mM TCEP) was used. Fractions were analysed by SDS–PAGE, and those containing pure octameric holo-complex were concentrated to around 5 μM and snap-frozen in aliquots for subsequent use.

**Pulldown experiments**

For interaction analyses between Nse5/6 complexes with Twin-Strep-tagged Smc5/6 hexamer, the Nse5/6 complexes (or Nse6N-CPD-His) were diluted to 1 μM in a total volume of 250 μl with buffer (10 mM Hepes-KOH pH 7.5, 150 mM potassium acetate, 2 mM MgCl$_2$, 20% glycerol) and an input sample was removed and boiled for 5 min after mixing with 2 × SDS gel-loading dye. Strep-Tactin Sepharose High Performance resin (GE Healthcare; 20 μl resin per pulldown) was incubated either with only 250 μl buffer or with the same buffer containing 0.5 μM Twin-Strep-tagged Smc5/6 hexamer. After 1-h incubation at 4°C on a rotating wheel, the resin was collected by centrifugation (2 min at 700 g) and washed twice with 500 μl of buffer to remove unbound hexamer. Pre-diluted Nse5/6 complexes were then incubated with either empty or loaded resin for 1 h at 4°C on a rotating wheel, and afterwards, the resin was collected by centrifugation and washed twice with 1 ml of buffer. Bound material was eluted with buffer supplemented with 2.5 mM desthiobiotin and analysed by SDS–PAGE.

Pulldowns between Smc6Δhinge(EQ)-3C-Twin-Strep / Nse4(N) and Smc5Δhinge(EQ) / Nse2 / Nse4can were carried out in a similar way, except that the indicated nucleotides were added to 2 mM final concentration during binding and washing.

**Limited proteolysis of the Nse5/6 complex**

The complex used for analysis by limited proteolysis was the Nse6/Nse5-3C-His(8) complex at a concentration of 14 mg/ml (120 μM). For the time course shown in Fig 2B, 19 μl of this protein was combined with 1 μl of trypsin (Sigma; 1 mg/ml) and the mixture was incubated at RT. At the indicated time points, 1-μl aliquots were removed, mixed with 1 × SDS loading dye and boiled for 5 min. All time points were then analysed by SDS–PAGE, and bands of interested were analysed by LC-MS/MS.

**LC-MS/MS analyses of Nse5/6 proteolytic fragments**

In-gel proteolytic cleavage with sequencing grade trypsin (Promega) was performed as described (Shevchenko et al, 2006). The peptides from the digestion were dried and redissolved in 0.05% trifluoroacetic acid, 2% acetonitrile for analysis with liquid chromatography coupled with tandem mass spectrometry. Samples were injected on a Q Exactive Plus mass spectrometer interfaced via a nano EASY-Spray source to an Ultimate 3000 RSLCnano HPLC system (Thermo Scientific). After loading onto a trapping microcolumn Acclaim PepMap100 C18 (20 mm × 100 μm ID, 5 μm, Thermo Scientific), peptides were separated on a reversed-phase Easy Spray C18 column

(50 cm × 75 μm ID, 2 μm, 100 Å, Thermo Scientific). A 4–76% acetonitrile gradient in 0.1% formic acid (total time 140 min) was used for the separation with a flow of 250 nl/min. Full MS survey scans were performed at 70,000 resolution. In data-dependent acquisition controlled by Xcalibur 4.0 software (Thermo Scientific), the 10 most intense multiply charged precursor ions detected in the full MS survey scan were selected for higher energy collision-induced dissociation (HCD, normalized collision energy NCE =27%) and analysed in the orbitrap at 17,500 resolution. The window for precursor isolation was of 1.5 *m/z* units around the precursor and selected fragments were excluded for 60 s from further analysis.

MS/MS data were analysed using Mascot 2.7 (Matrix Science, London, UK) set up to search the yeast proteome in the UniProt database (www.uniprot.org, reference proteome of *Saccharomyces cerevisiae (strain ATCC 204508 / S288c)*, January 2019 version: 6,049 sequences). Trypsin (cleavage at K, R) was used as the enzyme definition, allowing 2 missed cleavages. Mascot was searched with a parent ion tolerance of 10 ppm and a fragment ion mass tolerance of 0.02 Da. Iodoacetamide derivative of cysteine was specified in Mascot as a fixed modification. N-terminal acetylation of protein and oxidation of methionine were specified as variable modifications.

## Cross-linking mass spectrometry (XL-MS) experiments

### Cross-linking reactions

Protein samples were diluted with either ATPase buffer (10 mM Hepes-KOH pH 7.5, 150 mM potassium acetate, 20% glycerol, 2 mM $MgCl_2$) or salt buffer (20 mM Hepes-KOH pH 7.5, 250 mM NaCl) to a final concentration of 1.6 μM in 125 μl. For cross-linking in the presence of ligands, 2 mM ATP and 16 nM [plasmid]DNA (25 kbp) were added and the mixture incubated for 10 min at RT. This ratio of protein: [plasmid]DNA corresponds to around 250 bp dsDNA per complex and thus matches the conditions used in ATPase assays. PhoX cross-linker (5 mM stock in DMSO) was added to a final concentration of 0.25 mM and the mixture was incubated for 20 min at RT. The reactions were then quenched by addition of Tris–HCl pH7.5 (1 M stock) to a final concentration of 20 mM and samples were snap-frozen.

### Sample preparation

To denature the cross-linked proteins (125 μg in each sample), 4 M Urea and 50 mM Tris was added and the samples were ultrasonicated two times for 2 min with 0.5-s pulses (50% intensity) and 0.2-s pauses (SonoPuls, Bandelin). Next, 1 mM $MgCl_2$ and 1% benzonase was added and the mixture was incubated for 1 h at 37°C. For reduction and alkylation of the proteins, 40 mM 2-cloroacetamide (CAA, Sigma-Aldrich) and 10 mM tris(2-carboxyethyl)phosphine (TCEP; Thermo Fisher Scientific), and 100 mM Tris at pH 8.0 was added. After incubation for 20 min at 37°C, the samples were diluted 1:2 with MS grade water (VWR). Proteins were digested overnight at 37 °C by addition of 3 μg trypsin (Promega) and 2 μg LysC (Promega). After digestion, the solution was acidified with trifluoroacetic acid (TFA; Merck) to a final concentration of 1% and a pH of < 2. The peptide mixtures were purified via Sep-Pak $C_{18}$ 1cc vacuum cartridges (Waters) and the elution finally vacuum-dried.

Cross-linked peptides were enriched with Fe(III)-NTA cartridges (Agilent Technologies; Santa Clara, Ca) using the AssayMAP Bravo Platform (Agilent Technologies; Santa Clara, Ca) in an automated fashion (Post *et al*, 2017; Steigenberger *et al*, 2019). Cartridges were primed at a flow rate of 100 μl/min with 250 μl of priming buffer (0.1% TFA, 99.9% ACN) and equilibrated at a flow rate of 50 μl/min with 250 μl of loading buffer (0.1% TFA, 80% ACN). The flow-through was collected into a separate plate. Dried samples were dissolved in 200 μl of loading buffer and loaded at a flow rate of 5 μl/min onto the cartridge. Cartridges were washed with 250 μl of loading buffer at a flow rate of 20 μl/min and cross-linked peptides were eluted with 35 μl of 10% ammonia directly into 35 μl of 10% formic acid. Samples were dried down and stored at −20°C prior to further use. Before to LC–MS/MS analysis, the samples were resuspended in 0.1% formic acid.

### LC-MS/MS data acquisition

Enriched peptides were loaded onto a 30-cm analytical column (inner diameter: 75 μm; packed in-house with ReproSil-Pur $C_{18}$-AQ 1.9-micron beads, Dr. Maisch GmbH) by the Thermo Easy-nLC 1000 (Thermo Fisher Scientific) with buffer A (0.1% (v/v) Formic acid) at 400 nl/min. The analytical column was heated to 60°C. Using the nanoelectrospray interface, eluting peptides were sprayed into the benchtop Orbitrap Q Exactive HF (Thermo Fisher Scientific) (Scheltema *et al*, 2014; Hosp *et al*, 2015). As gradient, the following steps were programmed with increasing addition of buffer B (80% Acetonitrile, 0.1% Formic acid): linear increase from 8 to 30% over 60 min, followed by a linear increase to 60% over 5 min, a linear increase to 95% over the next 5 min and finally maintenance at 95% for another 5 min. The mass spectrometer was operated in data-dependent mode with survey scans from *m/z* 300–1,650 Th (resolution of 60k at *m/z* = 200 Th), and up to 15 of the most abundant precursors were selected and fragmented using stepped Higher-energy C-trap Dissociation (HCD with a normalized collision energy of value of 19, 27, 35) (Olsen *et al*, 2007). The MS2 spectra were recorded with dynamic *m/z* range (resolution of 30k at *m/z* = 200 Th). AGC target for MS1 and MS2 scans was set to 3E6 and 1E5, respectively, within a maximum injection time of 100 and 60 ms for the MS1 and MS2 scans, respectively. Charge state 2 was excluded from fragmentation to enrich the fragmentation scans for cross-linked peptide precursors.

### Data analysis

The acquired raw data were processed using Proteome Discoverer (version 2.5.0.400) with the XlinkX/PD nodes integrated (Klykov *et al*, 2018). To identify the cross-linked peptide pairs, a database search was performed against a FASTA containing the sequences of the proteins under investigation. Cysteine carbamidomethylation was set as fixed modification and methionine oxidation and protein N-term acetylation were set as dynamic modifications. Trypsin/P was specified as protease and up to two missed cleavages were allowed. Furthermore, identifications were only accepted with a minimal score of 40 and a minimal delta score of 4. Otherwise, standard settings were applied. Filtering at 1% false discovery rate (FDR) at peptide level was applied through the XlinkX Validator node with setting simple.

## Crystallization of the Nse6(177-C)/Nse5 complex

A selenomethionine substituted complex was prepared following a methionine biosynthesis feedback inhibition protocol (Burmann

*et al*, 2013) and concentrated to 18 mg/ml. Crystals were grown at 19°C by hanging-drop vapour diffusion from 2 µl drops formed by equal volumes of protein and of crystallization solution (12% (w/v) PEG 3350, 8% (v/v) 0.3 M Sodium malonate pH 7.5). Prior to flash freezing in liquid nitrogen, the crystals were briefly soaked in mother liquor containing 32% (v/v) ethylene glycol.

## Data collection and crystal structure determination

A single-wavelength anomalous diffraction experiment from selenium atoms (S-SAD) was performed at the macromolecular crystallography beamline X10SA (PXII) at the Swiss Light Source (Villigen, Switzerland). On a single crystal, a 360° data set was collected at 100 K at a wavelength of 0.9792 Å. The data analysis showed anisotropic diffraction in one direction. The data were processed using XDS and scaled and merged with XSCALE (Kabsch, 2010). Substructure determination and phasing were performed with SHELXC/D/E (Sheldrick, 2010) using the HKL2MAP interface (Pape & Schneider, 2004). The successful SHELXD substructure solution, in a search for 13 selenium sites, had a Ccall and a Ccweak of 49.9 and 22.5, respectively. Density modification resulted in a clear separation of hands. An initial model was built automatically with BUCCANEER (Cowtan, 2006). The model was completed by iterative cycles of model building in COOT (Emsley & Cowtan, 2004), followed by refinement in PHENIX (Adams *et al*, 2010). It is available at the Protein Data Bank under the accession code: 7OGG.

## Calculation of Nse5 and Nse6 surface conservation

Separate pdb-files for each protein were submitted to the ConSurf Server (Landau *et al*, 2005; Ashkenazy *et al*, 2016) and a multiple sequence alignment was generated automatically with standard settings, allowing max. 95% identity between sequences and min. 35% identity for homologs. The outputs were visualized in PyMol and colours were assigned to individual residues using the consurf_new script.

## Thermofluor buffer screening

The Smc5/6 hexamer showed a tendency to aggregate/precipitate in buffers with low ionic strength. For DNA-stimulated ATPase activity assays, we needed to find a buffer which is compatible with both protein stability and stable interaction with DNA substrates. As basic components, we chose 10 mM Hepes-KOH ph7.5, 2 mM MgCl$_2$ and 150 mM potassium acetate and used Thermofluor measurements in the presence of various additives such as detergents (NP40, DDM, Tween-20, TritonX-100), sugars (Glucose, Sucrose) or macromolecular crowding reagents (Glycerol, ethylene glycol, PEG 400). 2 µl protein solutions (0.1 mg/ml) and 18 µl of buffer containing 10X of Sypro Orange (Invitrogen) were added to the wells of a 96-well. The plate was sealed and heated in a real-time PCR system (Light Cycler 480, Roche diagnostic) from 20°C to 80°C in increments of 0.5°C/min. Fluorescence changes were monitored simultaneously. The wavelengths for excitation and emission were 498 and 610 nm, respectively. A simplified unfolding model (Chari *et al*, 2015) was used to fit the fluorescence data after normalization and obtain the temperature midpoint for the protein unfolding transition. The composition of the buffer chosen based on this analysis was 10 mM Hepes-KOH pH 7.5, 150 mM potassium acetate, 2 mM MgCl$_2$ and 20% glycerol (ATPase buffer).

## ATPase assays

ATPase activity measurements were done by a pyruvate kinase/lactate dehydrogenase coupled reaction at 25°C in ATPase buffer (10 mM Hepes-KOH pH 7.5, 150 mM potassium acetate, 2 mM MgCl$_2$, 20% glycerol). ADP accumulation was monitored for 1 h by measuring absorbance changes at 340 nm caused by NADH oxidation in a Synergy Neo Hybrid Multi-Mode Microplate reader. The 100 µl reactions contained 1 mM NADH, 3 mM phosphoenol pyruvic acid, 100 U pyruvate kinase, 20 U lactate dehydrogenase and the indicated concentrations of ATP and DNA substrates. The final concentration of Smc5/6 hexamer in the assay was 150 nM, except for the experiments with protein titration shown in Fig EV3B and C. The purified Twin-Strep-Nse6/Nse5-His(8) complex was added in two-fold molar excess to ensure complete formation of the octameric holo-complex. DNA substrates were added at a concentration corresponding to 250 bp DNA per complex. Results from these assays were analysed using the GraphPad Prism software.

## Fluorescence anisotropy measurements

Fluorescence anisotropy was measured using the same 40 bp dsDNA as the one used in ATPase assay, but with one strand modified at the 3' end with fluorescein. Measurements were recorded using a Synergy Neo Hybrid Multi-Mode Microplate reader (BioTek) with the appropriate filters in black 96-well flat bottom plates at 25°C. Buffer conditions were identical to those used in ATPase activity measurements. Anisotropy measurements were exported from the BioTek Synergy Neo software and subsequently fit to a binding polynomial using non-linear regression in GraphPad Prism 8.

## Analysis of salt-stable DNA association

100 µl reactions in ATPase buffer were set up containing combinations of the following components at the indicated concentrations: 600 nM of Smc5/6 hexameric complex (wild type or mutant), 900 nM of Nse5/6 complex, 2 mM nucleotide and 5 µg plasmid (pSG4418, 2.8 kbp, either circular or EcoRI-linearized). After incubation for 30 min at room temperature, 500 µl of ice-cold high-salt buffer (20 mM Tris pH 7.5, 1,000 mM NaCl, 5 mM EDTA) were added and the mixture was incubated with 20 µl of StrepTactin Sepharose HP (GE Healthcare) for 45 min to pull out proteins via a C-terminal Twin-Strep-tag on Smc6. ATPase buffer was instead used for control pulldowns (low salt). The bead suspensions were then transferred to Costar Spin-X centrifuge tube filters (0.45 µm cellulose acetate; Corning), and the liquid was removed from the beads by centrifugation for 2 min at 800 *g* at 4°C. The beads were washed 3 times with high-salt buffer, or with ATPase buffer as a control where indicated. Finally, the bound material was eluted with a buffer containing 20 mM Tris pH 7.5, 250 mM NaCl and 5 mM desthiobiotin. Aliquots of the eluate were either supplemented with 6 × gel-loading dye containing SDS (Thermo Scientific), heated to 65° for 10 min and analysed by agarose gel electrophoresis (1% in 0.5 × TBE) to check the DNA content, or supplemented with 2 × SDS gel-loading dye, heated to 95°C for 10 min and analysed by SDS–PAGE to visualize eluted proteins.

## Site-specific BMOE cross-linking

Smc5/6 hexamers with or without indicated reporter cysteines were diluted to a final concentration of 0.5 μM in ATPase buffer in a total volume of 40 μl. In reactions containing the Nse5/6 dimer, this complex was added in a 1.25 × molar excess (0.625 μM). Indicated reactions contained ATP (2 mM final concentration) and/or either [plasmid]DNA or [40bp]DNA. DNA substrates were always added at a concentration corresponding to around 250 bp per Smc5/6 complex, which corresponded to 5 nM of plasmid DNA and 3.2 μM of 40 bp oligonucleotides. Protein and substrates were incubated for 10 min at RT after mixing, and BMOE was then added at a final concentration of 1 mM. After 45 s incubation beta-mercaptoethanol was added at a concentration of 14 mM to stop the reaction. Samples were mixed with SDS gel-loading dye, heated to 80°C for 15 min and then analysed on Novex WedgeWell 4–12% Tris-Glycine Gels (Invitrogen). Gels were fixed for 1–2 h in gel fixing solution (50% ethanol, 10% acetic acid) and stained overnight using Coomassie staining solution (50% methanol, 10% acetic acid, 1 mg/ml Coomassie Brilliant Blue R-250). Gels were destained in destaining solution (50% methanol, 10% acetic acid) and then rehydrated and stored in 5% acetic acid. Quantification of bands in scanned gel images was done using Fiji (Schindelin *et al*, 2012).

## Electron microscopy

### Cryo-EM grid preparation and imaging

4 μl peak fractions at a final $Abs_{280nm}$ of ~0.3 (~1.3 μM) were applied to glow discharged ($2.2 \times 10^{-1}$ mbar for 20 s) Quantifoil holey carbon grids (R2/1, 200 mesh, Quantifoil). The grids were plunge vitrified into a liquid ethane/propane mix using a Vitrobot Mark IV at 4°C and 95% humidity. Cryo-EM data were collected on a FEI Talos Arctica microscope operated at 200 kV, equipped with a FEI Falcon 3EC direct detector operating in integrating mode. A total of 2,362 movies were recorded at a nominal magnification of 72,000× that corresponds to 1.997 Å/pixel at the specimen level using FEI EPU. The total exposure of 89.6 $e^-/Å^2$ at the specimen level was evenly distributed over 40 frames during 4 s. The preset target defocus range was 0.5–3.5 μm.

### Cryo-EM data processing

The RELION-3.0 implementation of MotionCor2 (Zheng *et al*, 2017) was used to correct for beam-induced sample motions and radiation damage. The summed and dose-weighted micrographs were used for further processing. Particles were selected using Gautomatch (https://www2.mrc-lmb.cam.ac.uk/download/gautomatch-056/). CTF parameters were determined using Gctf (Zhang, 2016). The 766170 initially picked particle candidates were subjected to several rounds of template-free, unsupervised 2D classification in RELION-3.0 (Kimanius *et al*, 2016; Zivanov *et al*, 2018). This resulted in the 2D class averages shown in Fig 1D.

## Data availability

The structural coordinates from this publication have been deposited to the Protein Data Bank (PDB) (www.rcsb.org) and assigned the identifier 7OGG (https://doi.org/10.2210/pdb7OGG/pdb). The

proteomics data have been deposited to the ProteomeXchange Consortium via the PRIDE partner repository with the dataset identifier PXD024160 (http://identifiers.org/px:PXD024160).

**Expanded View** for this article is available online.

## Acknowledgements

We thank members of the Gruber laboratory for comments on the manuscript, the Protein Analysis Facility (PAF) at FBM-UNIL for the identification of Nse5 and Nse6 fragments, the crystallization facility at the MPI of Biochemistry for assistance with crystal screening/optimization and the staff at the Swiss Light Source (SLS) for help with data collection. We are grateful to Serge Pelet and the Pelet laboratory for helpful advice and for sharing materials and reagents for genetic engineering in yeast. This work was supported by the Swiss National Science Foundation (310030L_170242), the European Research Council (Horizon 2020 ERC CoG 724482) to S.G and the German Research Foundation (DFG RA2941/1-1) to M.R. Additional support came from the Netherlands Organisation for Scientific Research (NWO) research programme TA with project number 741.018.201 and the European Union Horizon 2020 programme INFRAIA project Epic-XS (Project 823839) to R.A.S.

## Author contributions

MT cloned recombinant expression constructs and optimized protein expression and purification with assistance from Y-MS. MT purified all preparation of Smc5/6 and Nse5/6 and performed all biochemical assays. MT generated recombinant yeast strains. BS and RAS provided PhoX and analysed PhoX-cross-linked protein preparations. JB optimized crystallization conditions and collected X-ray diffraction data. JB built the Nse5/6 model with help from EL. CB optimized buffer conditions for purified protein preparations. IBS performed electron microscopy and reconstruction. MT and SG wrote the manuscript with input from all authors. MR and SG initiated the study.

## Conflict of interest

The authors declare that they have no conflict of interest.

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
