## [Review Process File · The EMBO Journal]

Nse5/6 inhibits the Smc5/6 ATPase and modulates DNA substrate binding

Michael Taschner, Jerome Basquin, Barbara Steigenberger, Ingmar Schafer, Claire Basquin, Esben Lorentzen, Markus Räschle, Richard Scheltema, and Stephan Gruber

DOI: [10.15252/emj.2021107807](https://doi.org/10.15252/emj.2021107807)

Corresponding author(s): *Stephan Gruber (stephan.gruber@unil.ch)*

Review Timeline:

Submission Date:	5th Feb 21
Editorial Decision:	12th Mar 21
Revision Received:	11th May 21
Editorial Decision:	7th Jun 21
Revision Received:	10th Jun 21
Accepted:	11th Jun 21

Editor: *Hartmut Vodermaier*

Transaction Report:

Thank you again for your patience during the external review of your manuscript on Smc5/6 complex architecture and function. We have now eventually received all three sets of reviewer comments, copied below for your information, and would in light of their overall interested and constructive feedback be happy to consider a revised manuscript further for publication. Before the paper should be acceptable for The EMBO Journal, there are however a number of substantive points raised in all three reports that would warrant careful addressing. As you will see, the referees note in particular that the structural and XL-MS analyses remain somewhat incomplete, as do follow-up mutational analyses and model preparation.

REFEREE REPORTS

Referee #1:

SMC complexes are key regulators of genome functions. The SMC5/6 complex is involved in DNA damage repair and also suppresses the transcription of viral genomes in mammals. The non-SMC components (Nse) of the SMC5/6 complex are very different from those in the well-studied cohesin and condensin complexes. In this manuscript, Gruber and his colleagues used biochemical and structural approaches to investigate the architecture of the budding yeast Smc5/6 core hexameric

complex and the octameric holo-complex that also contains Nse5/6. They found that the Nse5/6 subcomplex hinders head engagement of Smc5/6 ATPases to suppress the activity of the Smc5/6 core complex. Long DNA relieves this inhibition. They also determined the crystal structure of Nse5/6.

Together with several studies published recently, this work provides novel insights into the regulation of SMC complexes. The data presented are convincing. The writing is clear and logical. Publication in EMBO J is recommended, provided that the authors address the following points.

Major points:

1. The XL-MS data showed that, in the presence of ATP and plasmid DNA, the contacts between Nse5/6 and Nse1-4 are disrupted. The contacts between Smc5/6 and Nse3/4 also appear decreased. Is the Nse1-3-4 module still present in the complex? Can the authors use pull down experiments to verify that Nse1-3-4 remains bound to Smc5/6 under these conditions?
2. Why did the authors only collect cryo-EM data on the core hexameric complex, but not on the holo-complex? 2D averages of cryo-EM images of the octameric complex in the absence of ATP and DNA could lend strong support to the hypothesized 'inhibited conformation' of Smc5/6 holoenzyme.
3. It is interesting that the short DNA, but not long plasmid DNA, can stimulate Smc5/6 hexamer activity. Why can't the long DNA activate the Smc5/6 hexamer? In Fig. 5, long plasmid DNA decreases the CC-Cys crosslinking of the hexamer, but it does not affect head engagement even in the presence of ATP. What is the effect of short DNA on head engagement of the hexamer?
4. The authors show that ATP does not change the crosslinking of J, CC and E-Cys in vitro. This is rather surprising. It is possible that continuous ATP hydrolysis produces an ensemble of states, thereby affecting crosslinking efficiency. What happens if ATP is replaced by a nonhydrolyzable analogue?
5. The hinge has two interfaces. In human cohesin, both interfaces can be opened in the isolated hinge structures, while the north interface is opened in the structure of holo-complex. Which interface of the Smc5/6 hinge was chosen for the crosslinking experiment in Fig. 5B? Will the crosslinking of the other hinge interface be influenced by the addition of ATP and/or DNA?

Minor points:

1. The citations for the reference and figure in line 82 should be corrected.
2. In Fig. 5B-D, the MWs of the protein standard are missing.
3. In Fig. 6A, what do red dash lines mean?
4. The authors should tone down the statement in lines 325-326. It has been reported that the E. coli MukE inhibits the MukBF ATPase activity, and similarly, DNA alleviates this inhibition (Zawadzka, K., et al. eLife, 2018). This point might be discussed in the manuscript.

Referee #2:

The manuscript from Taschner et al., titled: "Nse5/6 inhibits the Smc5/6 ATPase to facilitate DNA substrate selection", describes architecture and biochemical features of the budding yeast SMC5/6 complex. It nicely fits to very recent "blast" of papers on the SMC5/6 topic. The authors focus on the structure and role of Nse5-Nse6 subcomplex. They bring new data on its inhibitory impact on SMC5/6 core activities. Particularly, they employ unique site-directed crosslinking approach to show different conformational states of SMC5/6 in the presence or absence of Nse5-Nse6. I believe that these new results will advance our understanding of the enigmatic SMC5/6 complex and I recommend them for publication after major revision.

Major comments:

1. The authors provide crystal structure of the Nse5-Nse6 dimer. However, they refrain from running mutagenesis to confirm their structural data (line 229). They should either perform their own control experiments (e.g. using Y2H system instead of in vitro expression) or refer to the mutagenesis data published in <https://www.biorxiv.org/content/10.1101/2020.12.31.424863v1> (these authors used in vitro expressed mutated proteins successfully).
2. The Nse6 N-terminal fragment (aa1-179) binds to the SMC5/6 hexamer and these amino acids are essential for yeast cell viability. In contrast, deletion of aa 1-86 is not lethal suggesting that these amino acids mediate only non-essential functions (e.g. binding to Rtt107).
 - a. The authors should prepare Nse6 (aa 86-179) fragment and show that aa 1-86 region is not essential for the Nse6 binding to SMC5/6 hexamer.
 - b. As SMC5/6 is involved in DNA repair, the "86-C" cells should be tested for their sensitivity to DNA-damaging agents (e.g. MMS, HU). The line 302 should be corrected accordingly.
3. It has been shown that SMC5/6 can bind different DNA substrates like ssDNA, dsDNA or scDNA (e.g. <https://pubmed.ncbi.nlm.nih.gov/25984708/> - paper should be cited accordingly). The authors show stimulation of ATPase activity by "circular" DNA and dsDNA.
 - a. Is the "circular" DNA nicked or in supercoiled form? It should be shown (e.g. on agarose gel) in Supplementary data.
 - b. Effect of ssDNA on ATPase activity should be shown
 - c. Binding of DNA substrates to SMC5/6 hexamer and octamer should be shown (e.g. by EMSA analysis)
4. The reason of the poor E-Cys crosslinking is not clear. Full evaluation of E-Cys crosslinking should be provided (E-Cys crosslinking efficiency in SMC5/EQ and SMC6/EQ mutants should be shown).

Minor comments:

1. The authors should deposit their crystal structure and crosslink data in appropriate databanks and made them available for reviewers. The data should be released for research community upon publication of the paper. Note, Nse6 amino acid ID numbers in the current PDB do not match their numeric positions in the protein sequence. In Fig. 2D, the Nse5-D45 residue is incorrectly displayed as E45.
2. The authors designed Cys-Cys mutations (for J-Cys and CC-Cys crosslinks) "based on models" of their previous SMC-ScpAB data (<https://pubmed.ncbi.nlm.nih.gov/28689660/>). Are these models compatible with their Lys-Lys crosslinking data? Modelling should be described in the manuscript.
3. The authors provide nice introduction to the SMC topic, however, they should improve referencing:

- a. following text needs careful editing: line 82 - "(refs and/or Fig. 1D?)" + line 365 - "(cit. Alt et al;".
- b. line 77 - DNA stimulation of the SMC5/6 complex was already shown here:
<https://pubmed.ncbi.nlm.nih.gov/10747036/>
- c. line 89 - Etheridge et al., 2020, does not show any data on „electron paramagnetic resonance and cross-linking"
- d. line 138 - primary reference for *S. pombe* Nse5/Nse6 non-essential function (<https://pubmed.ncbi.nlm.nih.gov/16478984/>) should be cited instead of and/or together with Oravcova et al., 2019; essential functions of the Nse5/Nse6 subunits were demonstrated only in yeast *S. cerevisiae* (<https://pubmed.ncbi.nlm.nih.gov/15738391/>), so far. The plant *A. thaliana* Nse5/SNI1 and NSE6/ASAP1 mutants are defective in root development, but they are not lethal (e.g. compared to *smc5* mutant; <https://pubmed.ncbi.nlm.nih.gov/24207055/>)
- e. line 464 - fission yeast NSE6 was mapped to arms by Palecek et al., 2006 (not by Pebernard et al., 2006); notably, Palecek et al. also showed that Nse5 binds to SMC5-SMC6 head constructs (but not to head-less fragments - supporting an essential role of head domains in these interactions).
- f. line 472 - equivalent interaction between human SMC5/6 and Slf2/Nse6 subunit has been mapped (while there is no evidence for the human Slf1/Nse5 subunit direct binding to SMC5-SMC6; Adamus et al., 2020).
- g. line 486 - the authors may consider other SUMO-related interactions of NSE5/NSE6 published in <https://pubmed.ncbi.nlm.nih.gov/14690591/>
- h. line 505 - "relocation" of the Nse1/3/4 module was also proposed for *S. pombe* SMC5/6 core complex in <https://pubmed.ncbi.nlm.nih.gov/32546830/>

4. Addition of Nse5-Nse6 to the SMC5/6 hexamer reduced J-Cys and CC-Cys crosslinking. The authors conclude that the binding of Nse5-Nse6 destabilizes the rod conformation (line 382). Alternative explanation considering masking effect of Nse5-Nse6 binding should be provided (given the position of Nse5-Nse6 within the complex, it may block access of the crosslinker to Cys residues).

5. Cloning details should be provided for the SMC5/6 constructs.

Referee #3:

Reviewer's comments on the manuscript by Taschner et al. "Nse/6 inhibits the ATPase to facilitate DNA substrate selection"

First of all, I would like to apologise for the untoward delay in reviewing this manuscript.

Taschner and colleagues have investigated how SMC complexes in yeast control DNA folding and topology.

The major findings of this work are:

1. The authors have solved the structure of the SMC subcomplex Nse5/6 by X ray crystallography.
2. They investigated the overall structure of the holoenzyme complex (consisting of eight different proteins) by crosslinking combined with mass spectrometry (XLMS) and obtained a 2D class average of the holoenzyme by cryo-EM.
3. They have mapped the contact sites of Nse5/6 subcomplex within the holoenzyme complex by XLMS.
4. They found that Nse5/5 inhibits the ATPase activity of SMC5/6.
5. They found that Nse5/6 induces a different conformation of the head domain of SMC5/6, which probably influenced the ATPase activity.

Overall, I like this work, as it adds novel insight into the molecular function of the SMC complex in

the light of binding of Nse5/6. The work is thus of potential interest for the cell-biology community. Nonetheless, I consider that the study can be significantly improved; especially, the documentation of some of the results is inadequate. I am also surprised that the authors do not provide an entire molecular model of the holoenzyme complex (taking account of their own and other available structural data); the XLMS, EM and crystallisation data would, in combination with e.g. molecular modelling and docking attempts, allow such a model to be constructed.

Specific points:

1. The authors have not provided any table of the crosslinked amino acids or of the peptide sequences for any of the XLMS experiments, and I cannot see any good reason for this omission. It is impossible to reconstruct the crosslinks shown in the various figures. I would have thought it was common sense to provide at least tables of crosslinked proteins and/or peptides. In this connection: An enigma remains in Figure 1c, 6a-c. The graphics and the figure legends do not explain what is shown in terms of protein domains. If in Figure 1c SMC5/6 domains are shown and in the current models both these proteins have a linear (with coiled-coil) shape structure, then it is unclear how to explain the observation that all the intra-crosslinks can reach from one (presumably the N terminal) end to the other (presumably C terminal) end. The statement in the Results section "The architecture of Smc5/6" that "The pattern of intralinks was generally in good agreement with available structural information" does not help much. The authors should state which "available structural information" they are referring to. And if such information is indeed available, then I would have appreciated it if the authors had mapped the distance constraints of their crosslinker to the distance of crosslinkable lysine residues. In my experience most of structural studies that had been published and made use of XLMS have provided such a plot.
2. The authors semi-quantified their crosslinks by counting identified crosslinks and observed changes in different states. I wonder whether a better statistic and better quantitative evaluation of the crosslinks can be performed; the authors fail to state whether they have related the number of sequenced crosslinks to the number of non-crosslinked peptides in the sample, in order to achieve a certain normalisation (and if not, then why not).
3. The authors used an enrichable and lysine-reactive protein crosslinker (named Phox) to map the overall structure of the holoenzyme. It is not clear why the authors did not use an additional crosslinker to obtain more crosslinking sites - e.g., also from different amino acids, which certainly would have extended the scope of their results. In this connection: the authors should offer an explanation for why the head domains of SMC5 and 6 do not crosslink to each other (Figure 1B).
4. The authors used a cysteine crosslinker to map conformational changes and visualised this by SDS-PAGE analyses. I am sure that this can - and should be - improved through the use of XLMS, monitoring the exact crosslinking sites. As only a restricted but much defined well-defined number of crosslinks can be expected in the in vitro system that they used, a targeted XLMS approach to monitor the crosslinked species at the peptide level would appear perfectly feasible.
5. The authors report that the ATPase activity of the SMC complex is altered upon binding of Nse5/6. The authors do not state whether (and if not, then why not) they also monitored ATP binding per se, e.g. by using a non-hydrolysable analogue. In other words, might ATP binding be impaired by the subcomplex Nse5/6?

We would like to thank the editor and all reviewers for the excellent reviewing of our manuscript. We appreciate all the supportive and constructive comments on the work. We believe that we have been able to significantly improve the manuscript by addressing these comments as described point-by-point below. Moreover, we now report experiments that demonstrate salt-stable binding of DNA by Smc5/6, briefly described below. We hope that the manuscript is now suitable for publication.

Based on our prior ATPase measurements, we have added new DNA association studies to the revised manuscript. As shown in Fig 4 (panel D to E), we found that salt-stable DNA association of Smc5/6 depends on ATP, on the Nse5/6 dimer, and on the circular nature of DNA, but not on ATP hydrolysis by Smc5/6 as previously claimed. We think these are worthwhile additions to the manuscript and great starting points for future investigations of the nature of Smc5/6 DNA association.

Referee #1:

SMC complexes are key regulators of genome functions. The SMC5/6 complex is involved in DNA damage repair and also suppresses the transcription of viral genomes in mammals. The non-SMC components (Nse) of the SMC5/6 complex are very different from those in the well-studied cohesin and condensin complexes. In this manuscript, Gruber and his colleagues used biochemical and structural approaches to investigate the architecture of the budding yeast Smc5/6 core hexameric complex and the octameric holo-complex that also contains Nse5/6. They found that the Nse5/6 subcomplex hinders head engagement of Smc5/6 ATPases to suppress the activity of the Smc5/6 core complex. Long DNA relieves this inhibition. They also determined the crystal structure of Nse5/6.

Together with several studies published recently, this work provides novel insights into the regulation of SMC complexes. The data presented are convincing. The writing is clear and logical. Publication in EMBO J is recommended, provided that the authors address the following points.

Many thanks for the support and appreciation of the work and for the valuable suggestions!

Major points:

1. The XL-MS data showed that, in the presence of ATP and plasmid DNA, the contacts between Nse5/6 and Nse1-4 are disrupted. The contacts between Smc5/6 and Nse3/4 also appear decreased. Is the Nse1-3-4 module still present in the complex? Can the authors use pull down experiments to verify that Nse1-3-4 remains bound to Smc5/6 under these conditions?

We have now performed pulldown experiments with all eight protein components either with or without ATP and plasmid DNA under the conditions used for XL-MS experiments. The new data (shown in Fig S6E) clearly shows that integrity of the octameric complex is not compromised in the presence of ATP and plasmid DNA and that the Nse4/3/1 module remains stably bound to the core complex.

2. Why did the authors only collect cryo-EM data on the core hexameric complex, but not on the holo-complex? 2D averages of cryo-EM images of the octameric complex in the absence of ATP and DNA could lend strong support to the hypothesized 'inhibited conformation' of Smc5/6 holoenzyme.

We have now also attempted to visualize the octameric complex by negative stain EM and cryo EM. Particles can clearly be deduced, however, further optimizations of these EM experiments (buffer screening, crosslinking, etc.) are needed to obtain higher-resolution reconstructions to reveal the exact position of the Nse5/6 module in the holoenzyme and to elucidate how Nse5/6 inhibits the ATPase. While our study was under review, a report by the Oliver and Murray labs was preprinted and then published (Hallet et al. 2021) describing medium-resolution structures of the octameric

Smc5/6 complex from *S. cerevisiae* using negative stain EM and 3D-reconstructions. The authors present evidence that Nse5/6 is wedged between Smc5 and Smc6 heads and accordingly might keep them apart. This finding is in agreement with our data. The study is now cited in our manuscript.

3. It is interesting that the short DNA, but not long plasmid DNA, can stimulate Smc5/6 hexamer activity. Why can't the long DNA activate the Smc5/6 hexamer? In Fig. 5, long plasmid DNA decreases the CC-Cys crosslinking of the hexamer, but it does not affect head engagement even in the presence of ATP. What is the effect of short DNA on head engagement of the hexamer?

The hexamer displays cooperative behaviour in the ATPase assay in the absence of DNA (Fig. S4B), meaning that the specific activity increases with elevating protein concentration. ^{40bp}DNA mildly stimulates (~2-fold) the ATPase regardless of protein concentration (Fig. S4B) due to a small but clear increase in *k_{cat}* (from about 40 per min to around 70 per min; see Fig 4B, right panel). In the presence of plasmid DNA, the main effect is the loss of cooperativity, presumably because hexamers bound to the long DNA molecules are isolated from one another. At low protein concentration, the activity is slightly higher (similar to the presence of short DNA), while at higher protein concentration the specific activity is apparently reduced, presumably due to the absence of cooperativity.

Regarding the effect of short DNA on head engagement, we have now carried out cysteine cross-linking experiments (with E-Cys) for hexamer and octamer (new Fig 5E). We found that the wild-type hexamer and octamer do not noticeably react to the short DNA substrate. Curiously, however, the hydrolysis mutant (double EQ) showed an increase in E-Cys cross-linking with short DNA in the hexamer (new Fig 5E). This effect is even more pronounced in the presence of Nse5/6. Thus, short DNA promotes head engagement in the EQ variant (which is intrinsically prone to engage heads) but not in wild-type complexes.

4. The authors show that ATP does not change the crosslinking of J, CC and E-Cys in vitro. This is rather surprising. It is possible that continuous ATP hydrolysis produces an ensemble of states, thereby affecting crosslinking efficiency. What happens if ATP is replaced by a nonhydrolyzable analogue?

Of note, we have previously observed limited effects of addition of ATP alone on the conformation of the bacterial Smc-ScpAB complex (e.g. by EPR, Vazquez Nunez, 2021).

We also failed to detect obvious effects when using ATPγS in place of ATP in the E-Cys cross-linking experiments (new Fig 5F). However, we believe ATPγS is not suitable for use with yeast Smc5/6 (and Smc-ScpAB) for reasons detailed further below. Our experiments with the hydrolysis-defective mutant complex (double EQ), however, supported the notion that head engagement is slow or thermodynamically unfavourable without DNA (Fig. 5E) even when ATP hydrolysis is hindered. It seems that head engagement is 'strictly' linked to DNA substrate binding at least in the octamer.

More in-depth analysis of ATPγS showed that it fails to promote head engagement to a level observed with ATP (as judged by pulldowns between Smc5(EQ) and Smc6(EQ) versions lacking the hinge dimerization domain; shown in new Fig S4F). ATPγS also failed to promote 'salt-stable DNA binding', a finding which has previously led to the claim that ATP hydrolysis is required to reach such a stable (potentially topological) DNA-bound state (Gutierrez-Escribano et al., 2020). We now show, however, that salt-stable binding is achieved by the hydrolysis-deficient EQ complex in the presence of ATP, providing further evidence that ATPγS is not a suitable analogue for use with the *S. cerevisiae* Smc5/6 complex. We believe that this is an important piece of information for the field, as at least two more groups have previously reported the use of ATPγS for several experiments with Smc5/6 (Gutierrez-Escribano et al, *Mol Cell*, 2020; Yu et al, *PNAS*, 2021) and other SMC complexes too.

5. The hinge has two interfaces. In human cohesin, both interfaces can be opened in the isolated hinge structures, while the north interface is opened in the structure of holo-complex. Which interface of the Smc5/6 hinge was chosen for the crosslinking experiment in Fig. 5B? Will the

crosslinking of the other hinge interface be influenced by the addition of ATP and/or DNA?

Many thanks for the suggestion. The interface we chose originally was the south interface, and this is now indicated in the new panels Fig 5B and S5A. We have also designed a cysteine pair for the north interface (Fig S5A), purified the resulting complex, and analysed it by crosslinking in the same way. The results are shown in Fig S5B. Overall, the crosslinking efficiency is slightly reduced at the north interface when compared to the south interface. More importantly, however, also new cysteine pair shows no significant difference in cross-linking in the presence of any combination of DNA, ATP, or the Nse5/6 complex. We thus conclude that any (transient or rare) opening/closing of this interface is not observable by our assay.

Minor points:

1. The citations for the reference and figure in line 82 should be corrected.

Many thanks! This has now been corrected.

2. In Fig. 5B-D, the MWs of the protein standard are missing.

In new Fig 5, all panels now include protein standards labels.

3. In Fig. 6A, what do red dash lines mean?

The red dash lines indicate cross-links that appear inconsistent with a fully extended conformation of the complex, thus indicating potential folding at various positions (or trans contacts between complexes). We now included a legend stating what these lines mean, both in Fig 6A, and in Fig 1C.

4. The authors should tone down the statement in lines 325-326. It has been reported that the E. coli MukE inhibits the MukBF ATPase activity, and similarly, DNA alleviates this inhibition (Zawadzka, K., et al. eLife, 2018). This point might be discussed in the manuscript.

Many thanks for bringing this to our attention. We are sorry for the oversight. We have moved this point to the discussion section where we cite the paper and provide a short discussion.

Referee #2:

The manuscript from Taschner et al., titled: "Nse5/6 inhibits the Smc5/6 ATPase to facilitate DNA substrate selection", describes architecture and biochemical features of the budding yeast SMC5/6 complex. It nicely fits to very recent "blast" of papers on the SMC5/6 topic. The authors focus on the structure and role of Nse5-Nse6 subcomplex. They bring new data on its inhibitory impact on SMC5/6 core activities. Particularly, they employ unique site-directed crosslinking approach to show different conformational states of SMC5/6 in the presence or absence of Nse5-Nse6. I believe that these new results will advance our understanding of the enigmatic SMC5/6 complex and I recommend them for publication after major revision.

Many thanks for the support and the constructive suggestions including the careful inspection of the references.

Major comments:

1. The authors provide crystal structure of the Nse5-Nse6 dimer. However, they refrain from running mutagenesis to confirm their structural data (line 229). They should either perform their own control experiments (e.g. using Y2H system instead of in vitro expression) or refer to the mutagenesis data published in <https://www.biorxiv.org/content/10.1101/2020.12.31.424863v1> (these authors used in vitro expressed mutated proteins successfully).

We had cited the Yu et al preprint in our initial submission with respect to their attempts to break up the Nse5/6 interface based on information obtained from their EM structure. We have updated the citation for the published paper during revision.

Considering the extensive shared interaction surface and the fact that neither Nse5 nor Nse6 can be produced in isolation using our system, interaction mutations are expected to compromise protein solubility and lead to failed protein production or purification. Related problems might limit the outcomes of Y2H experiments.

As an alternative and complementary strategy, we designed a cysteine pair predicted to be efficiently cross-linked with BMOE based on our structural model of Nse5/6. We selected H368C in Nse6 and G56C in Nse5 for two reasons: these residues are juxtaposed in closely interacting domains, and they show limited evolutionary conservation suggesting that the mutations should be well tolerated and not cause adverse effects on protein solubility/dimerization. After purification of the mutant complex we indeed detected robust (nearly complete) Nse5/6 crosslinking after only a 30 second incubation with BMOE, which we now show in Fig S2D. This finding together with the mutagenesis data by Yu et al. of residues (that are also predicted to be involved in dimerization based on our crystal structure) make us highly confident that the model reflects the actual structure of the Nse5/6 complex in solution.

2. The Nse6 N-terminal fragment (aa1-179) binds to the SMC5/6 hexamer and these amino acids are essential for yeast cell viability. In contrast, deletion of aa 1-86 is not lethal suggesting that these amino acids mediate only non-essential functions (e.g. binding to Rtt107).

a. The authors should prepare Nse6 (aa 86-179) fragment and show that aa 1-86 region is not essential for the Nse6 binding to SMC5/6 hexamer.

Many thanks! We have now cloned and purified the ScNse6(86-179)-CPDHis protein and show in the new panel Fig S3C that indeed it still interacts with the hexameric Smc5/6 complex based on pulldowns.

b. As SMC5/6 is involved in DNA repair, the "86-C" cells should be tested for their sensitivity to DNA-damaging agents (e.g. MMS, HU). The line 302 should be corrected accordingly.

As suggested, we now tested the (86-C) mutant for sensitivity to a variety of stress conditions (MMS, UV, HU) and compared it either to a original wild-type parental strain, or to a strain that had the same marker gene (*KanMX*) inserted downstream of the wild-type Nse6 locus (see new Fig S3D). To our surprise, we found that the nse6(86-C) mutant (lacking the previously documented Rtt107 interacting region) did not show any noticeable sensitivity to these treatments. To make sure that the treatments were working as intended, we made use of a recently described DNA damage-sensitive mutant, smc6(R135E) (Serrano et al, *Mol Cell* 2020). We indeed found that the mutant was highly sensitive under all these conditions. Possibly additional interactions between Rtt107 and Nse6 (or another component of the Smc5/6 complex) compensate for the loss of Rtt107/Nse6N interface. This finding will be of special interest to researchers working on the DNA damage aspects of Smc5/6.

3. It has been shown that SMC5/6 can bind different DNA substrates like ssDNA, dsDNA or scDNA (e.g. <https://pubmed.ncbi.nlm.nih.gov/25984708/> - paper should be cited accordingly). The authors show stimulation of ATPase activity by "circular" DNA and dsDNA.

a. Is the "circular" DNA nicked or in supercoiled form? It should be shown (e.g. on agarose gel) in Supplementary data.

We now cite the paper as suggested.

Our preparation of circular DNA was purified by maxi-prep, and agarose gel electrophoresis showed that it is a mixture of (mostly) supercoiled and (some) relaxed (presumably nicked) DNA. Because of the large plasmid size (25 kbp) the relaxed form does not properly enter the gel. Topo-I treatment of the plasmid as expected removed all traces of the supercoiled form and converted it into the relaxed form. This gel is now shown as Fig S4E and referenced accordingly in the main text.

b. Effect of ssDNA on ATPase activity should be shown

We have now performed ATPase assays with 40mer ssDNA showing that its effects are much less pronounced (when compared to dsDNA) for the hexamer and virtually absent for the octamer (see Fig S4D) despite clearly binding to both complexes as we now show by fluorescence anisotropy measurements. This is in agreement with a recently published study (Hallet et al, 2021). We now cite this newly published work in the revised manuscript.

c. Binding of DNA substrates to SMC5/6 hexamer and octamer should be shown (e.g. by EMSA analysis)

We now measured the affinity of hexamer, octamer, and Nse5/6 dimer for both 40 bp dsDNA and 40mer ssDNA by fluorescence anisotropy. The results are shown in Fig 4C. Nse5/6 alone did not detectably bind to any of the substrates, in agreement with two independent studies (Yu et al, 2021; Hallet et al, 2021) that are now cited accordingly. In contrast, the hexamer bound efficiently to both substrates, and Nse5/6 detectably increases this affinity in the context of the octamer, potentially by making the complex adapt a conformation favouring DNA interaction.

4. The reason of the poor E-Cys crosslinking is not clear. Full evaluation of E-Cys crosslinking should be provided (E-Cys crosslinking efficiency in SMC5/EQ and SMC6/EQ mutants should be shown).

We have now cloned, expressed, and purified the Smc5(EQ)/Smc6(EQ) hexamer and carried out E-Cys cross-linking under various conditions. The results are shown in Fig 5E. Curiously the ATP-hydrolysis-deficient complex showed very similar cross-linking efficiencies under most conditions. The only detectable difference is that cross-linking is increased by the presence of short DNA (^{40bp}DNA) both in the context of the hexamer and octamer, whereas this DNA substrate has no effect on the corresponding wild-type complexes. These results corroborate the view that head engagement is slow or thermodynamically unfavourable in the absence of suitable DNA substrates.

Minor comments:

1. The authors should deposit their crystal structure and crosslink data in appropriate databanks and made them available for reviewers. The data should be released for research community upon publication of the paper. Note, Nse6 amino acid ID numbers in the current PDB do not match their numeric positions in the protein sequence. In Fig. 2D, the Nse5-D45 residue is incorrectly displayed as E45.

Many thanks for pointing out the shortcomings. The discrepancies in the structural model have been corrected; the coordinates were deposited at the Protein Data Bank and will be available under the accession code: 7OGG.

2. The authors designed Cys-Cys mutations (for J-Cys and CC-Cys crosslinks) "based on models" of their previous SMC-ScpAB data (<https://pubmed.ncbi.nlm.nih.gov/28689660/>). Are these models compatible with their Lys-Lys crosslinking data? Modelling should be described in the manuscript.

Our 'modelling' approach is described in Fig S5C and simply consists of submitting the protein sequences to the Phyre2-server (using default settings) followed by superpositioning of the output models onto either the Smc rod-model (Diebold-Durand et al., 2017) or engaged head model (PDB: 5xg3). We believe that this short description is sufficient to allow for successful reproduction.

Concerning compatibility of these models with the lysine-lysine crosslinking data, we mapped the detected cross-links on structural models generated by Phyre2. Of a total of 33 cross-links that could be mapped on the models, 3 are above (albeit close) to the allowed distance constraint for PhoX. This indicates that the models represent an RMSD of at least 20 Å, but likely much better.

The indicated RMSD is not good enough to verify the folding of the proteins and , as we explained above, we would like to refrain from further modelling at this stage.

3. The authors provide nice introduction to the SMC topic, however, they should improve referencing:

a. following text needs careful editing: line 82 - "(refs and/or Fig. 1D?)" + line 365 - "(cit. Alt et al;".

Many thanks for pointing these mistakes out! They are now corrected in the revised version.

b. line 77 - DNA stimulation of the SMC5/6 complex was already shown here: <https://pubmed.ncbi.nlm.nih.gov/10747036/>

Thanks! This reference has now been included in the introduction.

c. line 89 - Etheridge et al., 2020, does not show any data on „electron paramagnetic resonance and cross-linking"

Sorry for this mistake. It has now corrected.

d. line 138 - primary reference for *S. pombe* Nse5/Nse6 non-essential function (<https://pubmed.ncbi.nlm.nih.gov/16478984/>) should be cited instead of and/or together with Oravcova et al., 2019; essential functions of the Nse5/Nse6 subunits were demonstrated only in yeast *S. cerevisiae* (<https://pubmed.ncbi.nlm.nih.gov/15738391/>), so far. The plant *A. thaliana* Nse5/SNI1 and NSE6/ASAP1 mutants are defective in root development, but they are not lethal (e.g. compared to *smc5* mutant; <https://pubmed.ncbi.nlm.nih.gov/24207055/>)

We have now cited the Pebernard et al paper for non-essential functions of pombe Nse5/6 and the Zhao and Bloble paper for the essential role of cerevisiae Nse5/6. Many thanks for the suggestion.

e. line 464 - fission yeast NSE6 was mapped to arms by Palecek et al., 2006 (not by Pebernard et al., 2006); notably, Palecek et al. also showed that Nse5 binds to SMC5-SMC6 head constructs (but not to head-less fragments - supporting an essential role of head domains in these interactions).

Thanks! This has been changed accordingly.

f. line 472 - equivalent interaction between human SMC5/6 and Slf2/Nse6 subunit has been mapped (while there is no evidence for the human Slf1/Nse5 subunit direct binding to SMC5-SMC6; Adamus et al., 2020).

Corrected.

g. line 486 - the authors may consider other SUMO-related interactions of NSE5/NSE6 published in <https://pubmed.ncbi.nlm.nih.gov/14690591/>

Thanks for the suggestion. However, since our work is not focused on SUMO-related functions of Smc5/6, we prefer to not add additional citations to the already long reference list.

h. line 505 - "relocation" of the Nse1/3/4 module was also proposed for *S. pombe* SMC5/6 core complex in <https://pubmed.ncbi.nlm.nih.gov/32546830/>

We have now included this citation in the 'relocation' statement.

4. Addition of Nse5-Nse6 to the SMC5/6 hexamer reduced J-Cys and CC-Cys crosslinking. The authors conclude that the binding of Nse5-Nse6 destabilizes the rod conformation (line 382). Alternative explanation considering masking effect of Nse5-Nse6 binding should be provided (given the position of Nse5-Nse6 within the complex, it may block access of the crosslinker to Cys residues).

Addition of Nse5/6 not only reduced J-Cys and CC-Cys crosslinking, but also E-Cys crosslinking. It is very unlikely that Nse5/6 masks all three cysteine pairs located at different positions within the head module simultaneously and efficiently enough to completely inhibit BMOE crosslinking in each case. Much more likely, these crosslinks are inhibited by Nse5/6 pushing the Smc5 and Smc6 heads apart. Further support for this comes from the recent paper (Hallett et al., 2021) which shows by negative staining/3D-reconstruction that Nse5/6 is wedged between the heads. This paper has now been cited accordingly.

5. Cloning details should be provided for the SMC5/6 constructs.

We now included further details on the design and the preparation of the protein expression constructs. This information should be sufficient to recapitulate and reproduce the cloning of all expression constructs. The sequence of the constructs will be available upon request (to ensure that the latest available version and information is shared).

Referee #3:

Reviewer's comments on the manuscript by Taschner et al. "Nse/6 inhibits the ATPase to facilitate DNA substrate selection"

First of all, I would like to apologise for the untoward delay in reviewing this manuscript.

Taschner and colleagues have investigated how SMC complexes in yeast control DNA folding and topology.

The major findings of this work are:

1. The authors have solved the structure of the SMC subcomplex Nse5/6 by X ray crystallography.
2. They investigated the overall structure of the holoenzyme complex (consisting of eight different proteins) by crosslinking combined with mass spectrometry (XLMS) and obtained a 2D class average of the holoenzyme by cryo-EM.
3. They have mapped the contact sites of Nse5/6 subcomplex within the holoenzyme complex by XLMS.
4. They found that Nse5/5 inhibits the ATPase activity of SMC5/6.
5. They found that Nse5/6 induces a different conformation of the head domain of SMC5/6, which probably influenced the ATPase activity.

Overall, I like this work, as it adds novel insight into the molecular function of the SMC complex in the light of binding of Nse5/6. The work is thus of potential interest for the cell-biology community. Nonetheless, I consider that the study can be significantly improved; especially, the documentation of some of the results is inadequate. I am also surprised that the authors do not provide an entire molecular model of the holoenzyme complex (taking account of their own and other available structural data); the XLMS, EM and crystallisation data would, in combination with e.g. molecular

modelling and docking attempts, allow such a model to be constructed.

We thank the reviewer for the great feedback and provide a point-by-point response below. As for modelling of the complex, we agree with the reviewer that at the very least large parts of the structure are known from similarity to other SMC complexes. However, the individual components of the complex have not been structurally resolved and would need to be predicted. At this point we feel structural modelling, although progressing at an excellent pace, is not yet sufficiently advanced to allow for building a reliable model based on the medium resolution XL-MS data and the very low-resolution EM hull.

Moreover, and more importantly, the XL-MS data likely represent multiple states of Smc5/6 that need to be discerned prior to model building; this could otherwise lead to false presumptions. For example, the contacts of Nse5/6 to the heads and to the coiled coils may stem from different sub-populations of complexes. Modelling of folded coiled coils to satisfy both sets of contacts (as in Yu et al., 2021), might thus be misleading. Therefore, we prefer not to build detailed models from the available information at this moment.

Specific points:

1. The authors have not provided any table of the crosslinked amino acids or of the peptide sequences for any of the XLMS experiments, and I cannot see any good reason for this omission. It is impossible to reconstruct the crosslinks shown in the various figures. I would have thought it was common sense to provide at least tables of crosslinked proteins and/or peptides. In this connection: An enigma remains in Figure 1c, 6a-c. The graphics and the figure legends do not explain what is shown in terms of protein domains. If in Figure 1c SMC5/6 domains are shown and in the current models both these proteins have a linear (with coiled-coil) shape structure, then it is unclear how to explain the observation that all the intra-crosslinks can reach from one (presumably the N terminal) end to the other (presumably C terminal) end. The statement in the Results section "The architecture of Smc5/6" that "The pattern of intralinks was generally in good agreement with available structural information" does not help much. The authors should state which "available structural information" they are referring to. And if such information is indeed available, then I would have appreciated it if the authors had mapped the distance constraints of their crosslinker to the distance of crosslinkable lysine residues. In my experience most of structural studies that had been published and made use of XLMS have provided such a plot.

We thank the reviewer for these excellent comments.

We note that all the relevant tables are accessible in the PRIDE repository. However, we do agree that it is helpful practice to provide key information on cross-linked peptide pairs also in the supplementary material. We thus now included fully annotated Crosslink Spectrum Match and Crosslink tables with the supplementary materials in the revised manuscript.

We apologize for not having properly labelled the domains in Fig 1 and Fig 6. This has now been corrected.

Regarding the fact that we detect intralinks from the N-terminal to the C-terminal end of the protein we failed to provide a clear explanation in the original version of the manuscript. In brief, SMC proteins fold at their central hinge-domain, bringing N- and C-terminal coiled-coil regions together to form long antiparallel coiled-coils. We now include a new panel in Fig S1C to show this schematically and refer to it in the text. While readers familiar with this protein family will know about this folding pattern, we agree that others may struggle to understand the crosslinks without this additional explanation.

2. The authors semi-quantified their crosslinks by counting identified crosslinks and observed

changes in different states. I wonder whether a better statistic and better quantitative evaluation of the crosslinks can be performed; the authors fail to state whether they have related the number of sequenced crosslinks to the number of non-crosslinked peptides in the sample, in order to achieve a certain normalisation (and if not, then why not).

Concerning the second point, it is exceedingly hard to normalize the intensities on the non-crosslinked peptides as these typically make up more than 99.9% of the sample (please refer to Steigenberger et al., 2019). To counter this, with PhoX we can enrich for the crosslinked peptides, which means we obtain two fractions (flow-through: non crosslinked peptides; eluate: crosslinked peptide pairs and monolinks) that need to be measured independently. In order to compare the different fractions, normalization needs to be applied (along the lines of e.g. MaxLFQ, which operates at the protein level), which uses information shared between the samples. As with our fractionation technique we separate peptides such normalization is not readily feasible.

3. The authors used an enrichable and lysine-reactive protein crosslinker (named Phox) to map the overall structure of the holoenzyme. It is not clear why the authors did not use an additional crosslinker to obtain more crosslinking sites - e.g., also from different amino acids, which certainly would have extended the scope of their results. In this connection: the authors should offer an explanation for why the head domains of SMC5 and 6 do not crosslink to each other (Figure 1B).

Of all available crosslinking reagents, over 90% exclusively uses NHS-ester chemistry which is the most efficient and successful strategy. Alternatively, there is the option to use zero-length reagents which still link a lysine residue to aspartic and glutamic acid. However, this will likely not provide much extra information, as these proteins are DNA binding proteins. This means they are extremely rich in lysines, which are spread out over the whole sequence. This means that we likely have captured most or all of the relevant information. For illustration, we add below a sequence coverage plot for Smc5 and Smc6 with highlighted monolinked lysine-residues - these are spread over the whole sequence (red and blue dots along the sequence; the light blue stretches denote detected peptides). This indicates that these residues are accessible to the linker and when the distance information can be captured, we likely would have obtained this.

The lack of cross-links between the head domains is indeed somewhat surprising in case of the Smc5/6 hexamer (as also seen in Yu et al., 2021). In case of the octamer, however, the lack of head-head cross-linking is expected from our data and the EM reconstructions (Hallet et al., 2021).

4. The authors used a cysteine crosslinker to map conformational changes and visualised this by SDS-PAGE analyses. I am sure that this can - and should be - improved through the use of XLMS, monitoring the exact crosslinking sites. As only a restricted but much defined well-defined number of crosslinks can be expected in the in vitro system that they used, a targeted XLMS approach to monitor the crosslinked species at the peptide level would appear perfectly feasible.

In contrast to lysine-lysine cross-linking, cysteine cross-linking is highly specific and in the ideal case targeted to a single pair of cysteines. For each of our cysteine-mutant complexes, we know the exact position of the generated cross-link and we can directly quantify the cross-linking efficiency by the detection of the cross-linked and the un-cross-linked fraction (by SDS-Page). Such relative measurements are quite precise and highly reproducible. We do not see the added value of using XLMS in this particular case as the structural information in this region is already known and this would solely represent a technical advance.

5. The authors report that the ATPase activity of the SMC complex is altered upon binding of Nse5/6. The authors do not state whether (and if not, then why not) they also monitored ATP binding per se, e.g. by using a non-hydrolysable analogue. In other words, might ATP binding be impaired by the subcomplex Nse5/6?

Our ATP titrations in the ATPase measurements indicate that the affinity of Smc5/6 for ATP is rather low, as also seen for many other ATPases. We estimated a K_m value for the hexamer of about 150 μM (Fig. 4) and expect that the K_d for ATP is in a similar range. In the octamer the K_m even increases to about 1000 μM . For reliable affinity measurements we would have to get the complexes concentrated to at least this concentration range, but we routinely only achieve stock concentrations of about 15 μM (corresponding to around 6 mg/ml).

Another potential complication is that the ATP binding pockets are completed only upon sandwiching of two ATP molecules between engaged Smc5 and Smc6 heads. Simple K_d measurements do not discriminate between ATP binding and ATP sandwiching. This means that defects in head engagement may also lead to apparently lowered affinity for ATP. We thus would not be able to differentiate if Nse5/6 hinders nucleotide binding to the head domains or their subsequent head engagement.

Nevertheless, we have attempted to determine K_d by different approaches: UV-cross-linking, DRaCALA and ITC. However, the obtained data are not convincing, and we would prefer not to include them in the manuscript.

We clearly state the two possibilities in our manuscript:

“This implies that Nse5/6 mainly inhibited the ATPase activity by precluding productive ATP binding by Smc5/6. It might do so by either interfering directly with the ATP binding step or by preventing subsequent head engagement, which completes the formation of the ATP binding pocket.”

While our work was under review/revision an independent study provided evidence by negative stain electron microscopy that the budding yeast Nse5/6 complex is wedged between Smc5 and Smc6 heads. We thus favour the scenario that the inhibition of ATPase activity is caused by steric hindrance of head-engagement in the presence of Nse5/6, and we cite this paper accordingly in our revised manuscript.

Thank you for submitting your revised manuscript to The EMBO Journal, and please excuse the slight delay in its reevaluation. We have now heard back from referee 2 who re-assessed the study, and following his/her positive comments (copied below), shall be happy to accept the study for publication, after incorporation of a few minor points raised by the referee, as well as the following editorial issues:

REFEREE REPORTS

Referee #2:

The revised manuscript from Taschner et al., titled: "Nse5/6 inhibits the Smc5/6 ATPase and modulates DNA substrate binding", describes the architecture and biochemical features of the budding yeast SMC5/6 complex. The authors addressed all the points raised in the review comments. They significantly improved their original manuscript making it ready for publication.

Only 2 minor mistakes to be rectified:

1. In Fig. 2D, the Nse5-D45 residue is still incorrectly displayed as E45.
2. Page 11, line 23: text "... or after linearization (Fig 4A, S4D)." shall refer to Fig. 4B

The authors should compare their crystal with the published PDB structure 7LTO and highlight differences between both structures.

Referee #2:

The revised manuscript from Taschner et al., titled: "Nse5/6 inhibits the Smc5/6 ATPase and modulates DNA substrate binding", describes the architecture and biochemical features of the budding yeast SMC5/6 complex. The authors addressed all the points raised in the review comments. They significantly improved their original manuscript making it ready for publication.

Only 2 minor mistakes to be rectified:

1. In Fig. 2D, the Nse5-D45 residue is still incorrectly displayed as E45.
2. Page 11, line 23: text "... or after linearization (Fig 4A, S4D)." shall refer to Fig. 4B

The authors should compare their crystal with the published PDB structure 7LTO and highlight differences between both structures.

We have corrected the two errors noted by reviewer #2 and as requested have added a superimposition of the recently published cryo-EM based model of Nse5/6 with our crystal structure (panel F in new Extended Data EV1) showing excellent agreement.

Thank you for submitting your final revised manuscript for our consideration. I am pleased to inform you that we have now accepted it for publication in The EMBO Journal.

Corresponding Author Name: Stephan Gruber

Manuscript Number: EMBOJ-2021-107807